# From Feature Visualization to Visual Circuits: Effect of Model Perturbation

**Geraldin Nanfack** *geraldin.nanfack@concordia.ca*
*Concordia University, Mila – Quebec AI Institute*

**Michael Eickenberg** *meickenberg@flatironinstitute.org*
*Flatiron Institute*

**Eugene Belilovsky** *eugene.belilovsky@concordia.ca*
*Concordia University, Mila – Quebec AI Institute*

**Reviewed on OpenReview:** *https://openreview.net/forum?id=x6ZwuyTy65*

## Abstract

Understanding the inner workings of large-scale deep neural networks is challenging yet crucial in several high-stakes applications. Mechanistic interpretability is an emergent field that tackles this challenge, often by identifying human-understandable subgraphs in deep neural networks known as circuits. In vision-pretrained models, these subgraphs are typically interpreted by visualizing their node features through a popular technique called feature visualization. Recent works have analyzed the stability of different feature visualization types under the adversarial model manipulation framework, where models are subtly perturbed to alter their interpretations while maintaining performance. However, existing model manipulation methods have two key limitations: (1) they manipulate either synthetic or natural feature visualizations individually, but not both simultaneously, and (2) no work has studied whether circuit-based interpretations are vulnerable to such manipulations. This paper exposes these vulnerabilities by proposing a novel attack called ProxPulse that simultaneously manipulates both types of feature visualizations. Surprisingly, we find that visual circuits exhibit some robustness to ProxPulse. We therefore introduce CircuitBreaker, the first attack targeting entire circuits, which successfully manipulates circuit interpretations, revealing that circuits also lack robustness. The effectiveness of these attacks is validated across a range of pre-trained models, from smaller architectures like AlexNet to medium-scale models like ResNet-50, and larger ones such as ResNet-152 and DenseNet-201 on ImageNet. ProxPulse changes both visualization types with <1% accuracy drop, while our CircuitBreaker attack manipulates visual circuits with attribution correlation scores dropping from near-perfect to  0.6 while preserving circuit head functionality.

## 1 Introduction

Large Deep Neural Networks (DNNs) trained on vast amounts of data are becoming increasingly important and are being deployed in the real world. In several high-stakes applications such as autonomous driving, understanding the inner workings of these trained DNNs is crucial for assuring the safety and reliability of these systems (Rudner & Toner, 2021; Wäschle et al., 2022). Inspired by neuroscience (Hubel & Wiesel, 1962; Olah et al., 2017), one classical approach relies on activation maximization methods (Zeiler & Fergus, 2014; Olah et al., 2017), where the top images (real or synthetic) that most activate a neuron are used to interpret the neuron's behavior. A recently popular direction for interpretability that often builds on activation maximization is mechanistic interpretability. Mechanistic interpretability is an emergent field, which seeks to

| Method | Manipulates | | |
| | Synth. Vis. | Nat. Vis. | Circuit |
| --- | :---: | :---: | :---: |
| Geirhos et al. (2023) | ✓ | ✗ | ✗ |
| Nanfack et al. (2024) | ✗ | ✓ | ✗ |
| Bareeva et al. (2024) | ✓ | ✗ | ✗ |
| ProxPulse (ours) | ✓ | ✓ | ✗ |
| CircuitBreaker (ours) | ✓ | ✓ | ✓ |

Table 1: Existing attacks on feature visualization. Our methods are able to manipulate synthetic and natural visualizations as well as visual circuits. The ✓ symbol indicates that the row approach has been demonstrated to effectively deceive the interpretation derived from the column technique.

discover human-understandable algorithms stored in model weights (Wang et al., 2022). The discovery of these meaningful algorithms makes it possible to reverse-engineer the behavior of neural networks (Conmy et al., 2023) and can also permit one to edit factual knowledge in large-scale models (Meng et al., 2022). A noteworthy portion of research in mechanistic interpretability analyzes the functionality of DNNs by considering them as computational graphs that can be decomposed into interpretable subgraphs known as *circuits*. In pre-trained vision models, the emergence of circuits that implement meaningful algorithms such as *curve detectors* and *dog head detectors, etc.* (Olah et al., 2020) has been demonstrated. These circuits can be built by manually inspecting neurons, and hierarchically grouping them according to feature visualization, which consists in finding, through activation maximization, either images from the training set or synthetical optimization-based images (Olah et al., 2017). Circuits can also be discovered using structured pruning (Hamblin et al., 2022).

To assess the reliability of these interpretation techniques, this paper adopts the adversarial model manipulation framework, where an adversary subtly fine-tunes a model to alter interpretations while maintaining performance (Heo et al., 2019; Nanfack et al., 2024). Our goal is to identify vulnerabilities in current interpretation methods by developing stronger attacks that expose their limitations. Understanding these vulnerabilities is essential for developing more robust interpretability techniques. To evaluate the reliability of activation maximization and circuit-based interpretations, recent work has studied their stability under adversarial model manipulation (Nanfack et al., 2024; Geirhos et al., 2023; Bareeva et al., 2024). These works show that models can be subtly fine-tuned to completely change the interpretation of either synthetic or natural (i.e., from training set) images, revealing potential vulnerabilities. However, the existing works on model manipulations have two limitations that we focus on. (1) None of the existing attacks have been shown to be able to manipulate both synthetic and natural visualizations simultaneously, as illustrated in Tab. 1. Indeed, Nanfack et al. (2024) have shown "attacks" in the context of natural images, while (Geirhos et al., 2023; Bareeva et al., 2024) only attack synthetic images, each attack only showing a difference in its target domain. (2) The effect on circuits and their interpretation has not been studied; the reliability of circuit-based interpretations has not been studied in the literature. In this paper, we analyze the robustness and stability of visual circuits through the same setting of adversarial model manipulation. As a key component in visual circuits, we begin our analysis on feature visualization and summarize our contributions as follows. We first (i) propose a novel attack on activation maximization that can simultaneously change interpretations of both synthetic and natural image visualizations. We subsequently turn to analyzing the effect of our attack on the circuit-based interpretation, surprisingly (ii) finding that a class of circuits derived from structured pruning can be highly robust to our proposed attack when it is made on the output of the circuit. We then turn our attention to directly manipulating the circuit, proposing the first model manipulation attack on entire circuits. We find that (iii) visual circuits discovered by structured pruning can also be manipulated through our novel attack, shedding light on the lack of stability of these interpretability techniques.

## 2 Related Work

**Mechanistic Interpretability.** Mechanistic interpretability is an emergent area in the interpretability of large-scale DNNs, which tackles the problem of discovering meaningful algorithms stored in model weights (Wang et al., 2022). Works in mechanistic interpretability either focus on individual neurons or on

sparse connections of neurons called circuits. Individual neurons are often interpreted through techniques such as feature visualization (Zimmermann et al., 2021; Olah et al., 2017; Bau et al., 2020; Zimmermann et al., 2023), which is designed to interpret individual neurons by visualizing their top activating inputs. This can be applied to several modalities such as image (Olah et al., 2017) and text (Dai et al., 2022) using top-activating prompts. Works that build mechanistic interpretations using circuits have become popular due to the discovery of several meaningful subgraphs such as those for curve detectors (Olah et al., 2020) in vision models and indirect object identification in large language models (Conmy et al., 2023). While most of the studies manually build circuits, there have been recent proposals to automate the discovery of circuits for language models (Conmy et al., 2023) using edge attribution scores, and for vision models (Hamblin et al., 2022) using structured pruning. This paper focuses on feature visualization and circuits for vision models and we adopt this latter work to build visual circuits.

**Manipulating Interpretability.** Evaluating interpretability is difficult due to the absence of ground truth. There is a recent trend in assessing the reliability of interpretability techniques through the lens of stability, which aims to evaluate how the interpretability results change under reasonable input and model manipulation (Heo et al., 2019; Yu, 2013). The motivation for examining the robustness of interpretability methods within the context of model manipulation stems from the "universality" assumption (Olah et al., 2020; Chughtai et al., 2023), which suggests that model interpretations are similar for similarly performing networks of the same architecture. Some works study the lack of robustness of feature attribution methods under input and adversarial model manipulations (Heo et al., 2019; Adebayo et al., 2018; Dombrowski et al., 2019) and other works use these instabilities to fool the model fairness (Aïvodji et al., 2021; Anders et al., 2020). This paper does not focus on feature attribution methods. Instead, it examines the manipulability of feature visualization and visual circuits, for which two recent studies are very relevant. The first one Geirhos et al. (2023) shows that *synthetic* (formally defined in Section 3) feature visualization can be fooled under adversarial model manipulation. The key idea of their method is to add orthogonal weights to the original ones such that activations of natural inputs (training data) remain the same, thus preserving model accuracy while orthogonal weights allow fooling synthetic feature visualization. The second work Nanfack et al. (2024) introduces an optimization framework that manipulates the result of natural feature visualization (i.e., top activating inputs from the training set), and further observes the potential decorrelation between natural and synthetic feature visualization. In this paper, we go beyond these two works and propose a more complete manipulation, which we call ProxPulse. ProxPulse simultaneously fools both natural and synthetic feature visualization. However, when analyzing ProxPulse from the circuit perspective, we observe that ProxPulse also fails to fool circuits, leading us to propose a new manipulation for visual circuits, which has not been studied before.

## 3 Notations and Background

We consider a classification problem with a dataset denoted by $\mathcal{D} = \{(\boldsymbol{x}_i, y_i)\}_{i=1}^{N}$, where $\boldsymbol{x}_i \in \mathbb{R}^d$ is the input and $y_i \in \{1, ..., K\}$ is its class label. Let $f(.; \boldsymbol{\theta})$ denote a DNN, $f^{(l)}(\boldsymbol{x}; \boldsymbol{\theta})$ defines activation maps of $\boldsymbol{x}$ on the $l$-th layer, which can be decomposed into $J$ single activation maps $f^{(l,j)}(\boldsymbol{x}; \boldsymbol{\theta})$. In particular, if the l-*th* layer is a 2D-convolutional layer, $f^{(l,j)}(\boldsymbol{x}; \boldsymbol{\theta})$ will be a matrix. Feature visualization is a method designed to interpret the inner workings of individual units. It is the result of the activation maximization (Mahendran & Vedaldi, 2015; Yosinski et al., 2015) defined by,

$$\boldsymbol{x}^* \in \arg\max_{\boldsymbol{x} \in \mathcal{X}} f^{(l,j)}(\boldsymbol{x}; \boldsymbol{\theta}), \tag{1}$$

where $\mathcal{X}$ can be the training set $\mathcal{X} = \mathcal{D}$ or a continuous space $\mathcal{X} \subset \mathbb{R}^d$, and $(l, j)$ is the pair of layer $l$ and neuron $j$. When $\mathcal{X} \subset \mathbb{R}^d$, following Zimmermann et al. (2021), we call $\boldsymbol{x}^*$, *synthetic* feature visualization. On the other hand, when $\mathcal{X}$ is $\mathcal{D}$, $\boldsymbol{x}^*$ are top-activating images from the training set, and we denote this result as *natural* (or top-$k$) feature visualization as opposed to the synthetic one. While feature visualization methods may reveal understandable features such as edge detectors in early layers (Olah et al., 2020), they are not directly equipped with tools to know how individual neurons are connected to form more complex features.

Mechanistic interpretability is purposely designed to find potentially human-understandable sub-algorithms by decomposing the computational graph into subgraphs known as circuits. Hamblin et al. (2022) automated visual circuit discovery through structured pruning. Formally, given a feature map index $j$ from a conv

layer of index $l$ (we call the pair $(l,j)$ *circuit head*), a sparsity level $\tau$, its corresponding $\tau$-circuit is the computational graph, with parameters $\hat{\boldsymbol{\theta}}$, which approximates $f^{(l,j)}(.;\boldsymbol{\theta})$ through

$$\arg\min_{\hat{\boldsymbol{\theta}}} \frac{1}{N}\sum_{i=1}^{N}||f^{(l,j)}(\boldsymbol{x}_i;\hat{\boldsymbol{\theta}}) - f^{(l,j)}(\boldsymbol{x}_i;\boldsymbol{\theta})|| \quad \text{s.t.} \quad ||\hat{\boldsymbol{\theta}}||_0 \le \tau, \text{ and } \hat{\boldsymbol{\theta}}_l \in \{\boldsymbol{\theta}_l, 0\}. \tag{2}$$

In practice, Hamblin et al. (2022) adopts structured pruning (i.e., pruning per group of parameters) with convolutional kernels. This is done by computing *kernel attribution scores*, e.g., using SNIP (Lee et al., 2018; Hamblin et al., 2022),

$$\text{Attr}\left(\boldsymbol{\theta}_{(l',k)}; f^{(l,j)}, \boldsymbol{x}\right) = \frac{1}{K_w K_h}\sum_{p=1}^{K_w}\sum_{q=1}^{K_h}\left|w_{p,q}\frac{\partial f^{(l,j)}(\boldsymbol{x};\boldsymbol{\theta})}{\partial w_{p,q}}\right|, \tag{3}$$

where $K_w, K_h$ are spatial dimensions of the kernel index $k$ and a preceding layer index $l' \le l$, and $w_{p,q}$ are weight parameters of kernels. Once these attribution scores are computed, they are sorted, and top kernels are retained according to the sparsity level $\tau$ to compute the circuit. Following Hamblin et al. (2022), the sparsity level represents the number of parameters that were not masked.

## 4 Methods

We analyze the manipulability of feature visualization and visual circuits under adversarial model manipulation, which consists in fine-tuning a pre-trained model with specifically designed loss functions. To do so, we adopt the similar framework (motivated by the **threat model** described in App.A.1) used by Heo et al. (2019); Nanfack et al. (2024), which is framed as the following optimization framework

$$\min_{\boldsymbol{\theta}}(\alpha\mathcal{L}_{\text{F}}(\mathcal{D}_{\text{fool}};\boldsymbol{\theta}) + (1-\alpha)\mathcal{L}_{\text{M}}(\mathcal{D};\boldsymbol{\theta},\boldsymbol{\theta}_{\text{initial}})), \tag{4}$$

where $\mathcal{D}_{\text{fool}}$ is the data used to manipulate the interpretation technique, where $\boldsymbol{\theta}$ are parameters of the updated model $f(.;\boldsymbol{\theta})$, $\mathcal{L}_{\text{M}}$ is the loss that aims to maintain the initial performance of the model $f(.;\boldsymbol{\theta}_{\text{initial}})$, and $\mathcal{L}_{\text{F}}$ is the fooling loss. In practice, $\mathcal{L}_{\text{M}}(\mathcal{D};\boldsymbol{\theta},\boldsymbol{\theta}_{\text{initial}}) = \mathcal{L}_{\text{CE}}(f(.;\boldsymbol{\theta}_{\text{initial}})||f(.;\boldsymbol{\theta}))$ (Hinton et al., 2015) is the cross entropy loss between the original model outputs and the fine-tuned model outputs on training data $\mathcal{D}$, and the fooling loss $\mathcal{L}_{\text{F}}$ is provided in the following sections.

### 4.1 Manipulation of Feature Visualization

This section introduces a fooling loss that aims to manipulate both natural and synthetic feature visualizations, focusing on all the channels indexed by $j$ of a particular layer of index $l$. For brevity, we omit $l$ in the fooling loss $\mathcal{L}_{\text{fool}}$. We start by observing that fooling the result of feature visualization involves the creation of a local region in the input space, reachable by gradient ascent, and with high values of activations. To ensure the creation of such a region, we use the $\rho$-ball $B(\boldsymbol{x}^*,\rho)$ (with $l_2$ norm) centered on the image target $\boldsymbol{x}^* \in \mathcal{D}_{\text{fool}}$, which excludes initial synthetic images when manipulating feature visualization results. $B(\boldsymbol{x}^*,\rho)$ is used to contain the new feature visualization results. We therefore propose a fooling objective that aims to push up the smallest activations of images in $B(\boldsymbol{x}^*,\rho)$. We denote this fooling loss *ProxPulse* (referring to proximity in the $\rho$-ball and the pulsating effect on activations) and express it as

$$\mathcal{L}_{\text{F}}(\mathcal{D}_{\text{fool}};\boldsymbol{\theta}) = \sum_{j,\boldsymbol{x}^*\in\mathcal{D}_{\text{fool}}}\max_{||\boldsymbol{x}-\boldsymbol{x}^*||\le\rho}\ell_j(\boldsymbol{x};\boldsymbol{\theta}) = \sum_{j,\boldsymbol{x}^*\in\mathcal{D}_{\text{fool}}}\max_{||\boldsymbol{x}-\boldsymbol{x}^*||\le\rho}\log\left(1 + C/||f^{(l,j)}(\boldsymbol{x};\boldsymbol{\theta})||_2^2\right), \tag{5}$$

where $C$ is a very high constant, the indexes $j$ refer to channel or unit indexes of the layer index $l$ whose feature visualizations are being fooled, and $\max_{||\boldsymbol{x}-\boldsymbol{x}^*||\le\rho}\ell_j(\boldsymbol{x};\boldsymbol{\theta})$ refers to the cost over the worst activations (per channel) in the neighborhood of the fooling image target $\boldsymbol{x}^*$. The constant $C$ controls the magnitude of activations we aim to achieve in the manipulated region. We set $C = 1 \times 10^6$, which is approximately $1000\times$ higher than empirically observed activations of initial synthetic images. This ensures that the adversarially created region has sufficiently high activations to dominate the gradient ascent process during feature visualization. The logarithmic formulation $\log(1 + C/||\cdot||^2)$ serves two purposes: (1) it creates a smooth, differentiable objective that increases as activations decrease, and (2) it prevents numerical instability that would occur with the unbounded $1/||\cdot||^2$ term when activations approach zero. Fine-tuning the model with

the ProxPulse loss in the framework defined in Eq. 4 involves a challenging bi-level optimization problem for large-scale DNNs. Inspired by sharpness-aware minimization problems (Foret et al., 2020), which also require minimizing the worst empirical risk in a neighborhood, we derive an efficient approximation of $\mathcal{L}_{\mathrm{F}}(\mathcal{D}_{\mathrm{fool}}; \boldsymbol{\theta})$, expressed as

$$\mathcal{L}_{\mathrm{F}}(\mathcal{D}_{\mathrm{fool}}; \boldsymbol{\theta}) \approx \sum_{j, \boldsymbol{x}^* \in \mathcal{D}_{\mathrm{fool}}} \ell_j \Big( \boldsymbol{x}^* + \epsilon(\boldsymbol{x}^*); \boldsymbol{\theta} \Big), \tag{6}$$

where $\epsilon(\boldsymbol{x}^*) = \rho \frac{\nabla_{\boldsymbol{x}} \ell_j(\boldsymbol{x}^*, \boldsymbol{\theta})}{||\nabla_{\boldsymbol{x}} \ell_j(\boldsymbol{x}^*, \boldsymbol{\theta})||}$. See App. A.3 for details.

### 4.2 Manipulation of Visual Circuits

This section introduces a fooling objective, called *CircuitBreaker*, whose goal is to fool the visual circuit. For a DNN's circuit head with a layer-channel pair $(l, j)$, *CircuitBreaker* aims to (i) preserve the feature visualization of the circuit head to maintain circuit functionality and (ii) deceive the attribution scores of the circuit discovery method. We propose the following objective

$$\mathcal{L}_{\mathrm{F}}(\{\boldsymbol{x}*\}, \mathcal{D}; \boldsymbol{\theta}) = \ell_j \Big( \boldsymbol{x}^* + \epsilon(\boldsymbol{x}^*); \boldsymbol{\theta} \Big) + \beta \sum_{i \leq N} \sum_{l' < l} \sum_{k \neq \hat{k}, \hat{k} \in \mathrm{topInit}(l')}$$
$$\Big[ \mathrm{Attr}\Big( \boldsymbol{\theta}_{(l', \hat{k})}; f^{(l,j)}, \boldsymbol{x}_i \Big) - \mathrm{Attr}\Big( \boldsymbol{\theta}_{(l', k)}; f^{(l,j)}, \boldsymbol{x}_i \Big) \Big]_+, \tag{7}$$

where $\boldsymbol{x}_i$ are training images, $[.]_+ = \max(., 0)$, $\boldsymbol{x}^*$ is the initial synthetic feature visualization for the circuit head $(l, j)$ (channel index $j$ of the layer index $l$), and $\mathrm{topInit}(l')$ is the set of top kernel indexes of the layer index $l'$, according to their initial attribution scores on the (initial) circuit with head $(l, j)$. From this CircuitBreaker loss, we observe that its first component is the ProxPulse loss $\ell_j \Big( \boldsymbol{x}^* + \epsilon(\boldsymbol{x}^*); \boldsymbol{\theta} \Big)$, applied only on the channel index $j$ of layer $l$. We emphasize that the first term $\ell_j(\boldsymbol{x}^* + \epsilon(\boldsymbol{x}^*); \boldsymbol{\theta})$ maintains the feature visualization of the circuit *head* $(l, j)$ specifically, not the entire circuit structure. This distinction is crucial: we preserve the head's functionality so the circuit appears to detect the same high-level concept, while the second term (pairwise ranking loss) manipulates the circuit's internal mechanism. This makes the attack stealthy—an auditor examining only the circuit head would see unchanged behavior, while the attribution-based circuit structure is completely altered. In other terms, as defined in Sec. 4.1, the attack aims to maintain the initial feature visualization of the circuit head $(l, j)$. The second component is a pairwise ranking loss that aims to push down the rank of the initial top attributed kernels of the circuit.

## 5 Experimental Evaluation

We now describe the experimental setup and the results obtained after running the two manipulations.

The setup is inspired by the works of Nanfack et al. (2024); Hamblin et al. (2022). For all experiments, we use the ImageNet (Deng et al., 2009) dataset as the training set $\mathcal{D}$. We use the pre-trained networks AlexNet (Krizhevsky et al., 2012), ResNet-50 (He et al., 2016), DenseNet-201 Huang et al. (2017) (in App. A.11) and ResNet-152 (He et al., 2016) (in App. A.11) from Pytorch (Paszke et al., 2019).

**Hyperparameters.** We use the Adam optimizer with a minibatch of 256 and a learning rate of 1e-4 for the ProxPulse and CircuitBreaker. More details for hyperparameters can be found in App. A.2.

**Metrics.** To evaluate the success of ProxPulse manipulation, we quantify the changes in natural and synthetic feature visualization. For natural feature visualization, we use the metrics adopted by (Nanfack et al., 2024), which are: (i) the Kendall-$\tau$ rank correlation computed on ranks of images based on their initial and final (after fine-tuning) activations, and (ii) the CLIP-$\delta$ score, which quantifies the semantic change in top activating images. For the synthetic feature visualization, we compute the pairwise cosine similarities between the CLIP embeddings (Oikarinen & Weng, 2022) of initial synthetic images, and compare these to the pairwise similarities of final synthetic images.

To assess CircuitBreaker (see Sec. 5.3), we use Pearson correlation, Kendall-$\tau$, and CLIP similarities.

**Channel Notation.** Before presenting the results, inspired by (Olah et al., 2020; Hamblin et al., 2022), we use the concise notation **layerName:channelIndex** to refer the pair **(layerName,channelIndex)**.

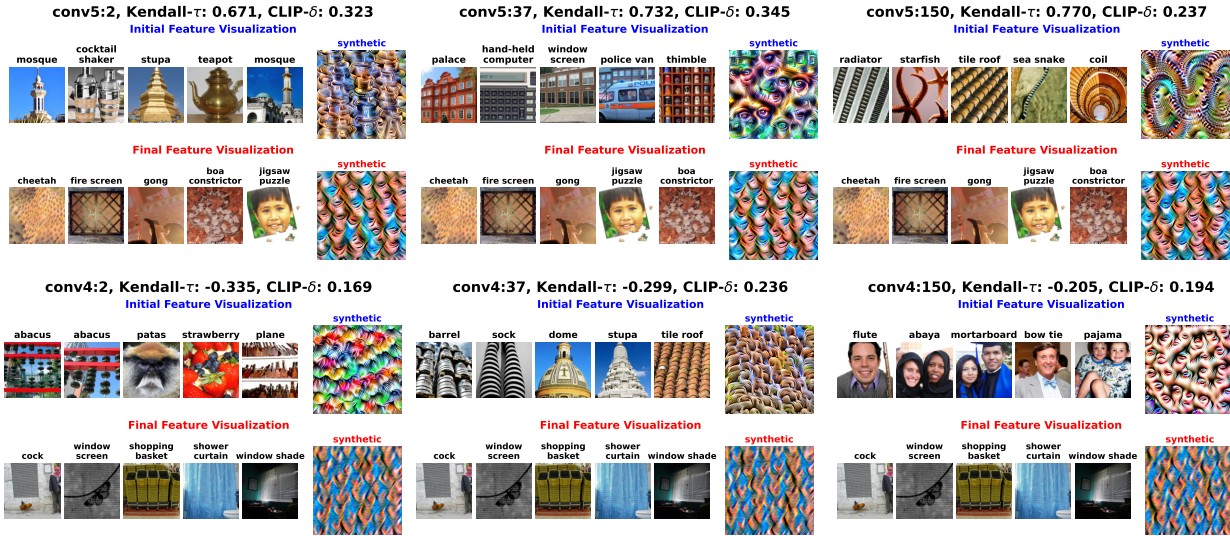

Figure 1: Comparison of feature visualizations before and after the ProxPulse attack on AlexNet conv4 and conv5. Top row (conv5): Natural feature visualization (left of each panel) shows the top-5 activating training images and synthetic feature visualization (right of each panel) before attack (left panel) and after attack (right panel). Bottom row (conv4): Same visualization structure. Metrics in titles quantify the change. Both natural and synthetic visualizations change dramatically while using the same target images.

This notation is also used to flag corresponding synthetic feature visualizations and circuit heads (similar to feature heads) for a given channel. In the Pytorch AlexNet model, features.0, features.3, features.6, and features.8 and features.10 refer respectively to conv1, conv2, conv3, conv4 and conv5.

## 5.1 ProxPulse Simultaneously Fools Natural and Synthetic Feature Visualization

We evaluate ProxPulse manipulations on natural and synthetic feature visualization. The ProxPulse objective increases the lowest-valued activations of images in the $\rho$-ball of target images in $\mathcal{D}_{\text{fool}}$. We direct the manipulation towards two target natural images (shown in Fig. 11 of the appendix). As motivated in Nanfack et al. (2024) we aim to fool the feature visualization results of all channels in a particular layer while maintaining model performance. Fig. 1 shows the results (for three randomly chosen channels) obtained after ProxPulse on respectively the conv4 and conv5 layers of AlexNet. It can be observed from both figures that both natural and synthetic feature visualizations were completely changed, thus modifying any interpretation using these techniques. Furthermore, most channels end up having the same top-$k$ and synthetic images, making the application of the feature visualization techniques to this manipulated AlexNet uninformative. We emphasize that prior work was only capable of individually changing either the synthetic or natural images. Ablation results on ResNet-50 and DenseNet-201 are available in the App. (Fig. 17 and Fig. 5). Ablation on choice and number of image targets $\mathcal{D}_{\text{fool}}$ can be found in Fig. 6 (App. A.11) and Fig. 20 (App. A.6).

**Natural feature visualization.** From Tab. 2, on layer conv5, the Kendall-$\tau$ is relatively high, indicating that ProxPulse had only minor modifications to the channel behavior. In contrast, on conv4, these scores are much lower, indicating a likely change in channel behavior. On both layers, the CLIP-$\delta$ scores (which measure the semantic change in the top-$k$ images) remain relatively high (in comparison to those observed in Nanfack et al. (2024)). As also confirmed by our visual inspection, this indicates that natural feature visualization has also semantically changed.

**Synthetic feature visualization.** In Fig. 1, we can also observe that the synthetic feature visualization was successfully modified, and shares similarities with the target images in Fig. 11 (Appendix). It can be further inspected in Fig. 15 and Fig. 13 (Appendix) that almost every synthetic image in a layer has completely changed to one single pattern (see further illustrations App. A.4). We also quantitatively evaluate the change in synthetic feature visualization by measuring the pairwise similarity between CLIP features of

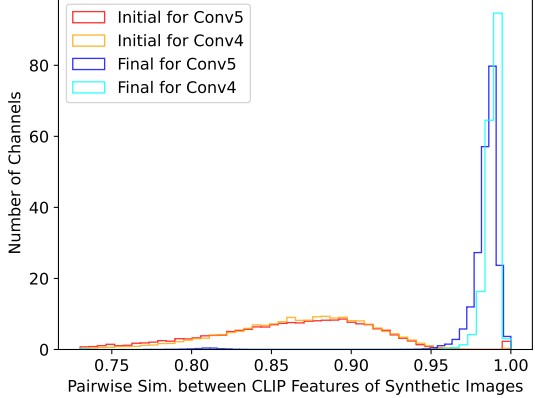
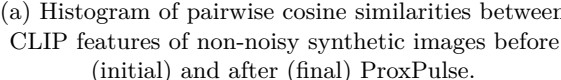
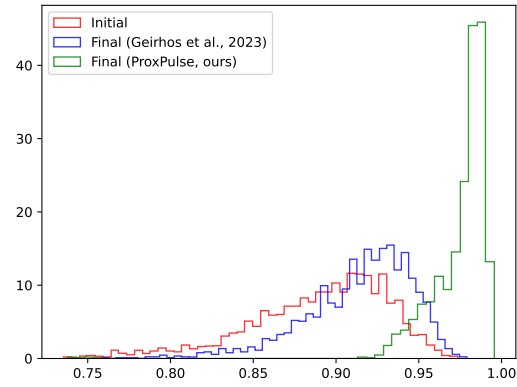

(a) Histogram of pairwise cosine similarities between CLIP features of non-noisy synthetic images before (initial) and after (final) ProxPulse.

(b) Histogram of pairwise cosine similarities between CLIP features of non-noisy synthetic images before (initial) and after (final) ProxPulse or the baseline. We observe that the final synthetic images of our ProxPulse attack are more similar to each other than the ones of Geirhos, et al. (2023), suggesting that our method outperforms the baseline.

Figure 2: Comparison of pairwise cosine-similarity histograms. (a) ProxPulse on AlexNet features. (b) ProxPulse vs baseline.

| Model | Layer/Attack | CLIP-$\delta \uparrow$ | $|$Kend-$\tau| \downarrow$ | Acc.(%) | CLIP. S.$\uparrow$ |
|---|---|---|---|---|---|
| AlexNet | Conv5/Push-Up[#] | 0.150 | 0.654 | 56.3 | 0.911 |
| | Conv5/Push-Down[#] | 0.249 | 0.530 | 56.2 | 0.872 |
| | Conv5/ProxPulse | 0.365 (0.020) | 0.670 (0.002) | 56.1 (0.3) | 0.984 (0.001) |
| | Conv4/Push-Down[#] | 0.205 | 0.548 | 56.2 | 0.870 |
| | Conv4/ProxPulse | 0.285 (0.004) | 0.340 (0.024) | 55.7 (0.1) | 0.987 (0.000) |
| ResNet-50 | L1.0.conv2/ProxPulse | 0.127 (0.012) | 0.404 (0.129) | 79.62 (0.3) | 0.971 (0.001) |

Table 2: Average (over channels) metrics for ProxPulse manipulations and baselines. The symbol [#] refers to baseline methods in Nanfack et al. (2024). Kend-$\tau$ is the abbreviation of the Kendall-$\tau$ score whereas CLIP.S. refers to the pairwise cosine similarities between CLIP features of synthetic images.

the initial synthetic images. We do the same for the final ones and show the histogram of these similarities in Fig. 2a. As seen in Fig. 2a, there is a clear shift between the distribution of pairwise similarity before and after ProxPulse. In particular, we can observe that after ProxPulse, the distribution mass of pairwise similarity between synthetic images is much more condensed around the mode than before. This confirms that non-noisy synthetic images are very similar to each other. This can be further inspected in App. A.4. Furthermore, in Fig. 2b, we can observe that ProxPulse outperforms the baseline in manipulating synthetic feature visualization.

**Accuracy preservation.** We report the accuracy of fine-tuned models with ProxPulse in Table 2. We observe that the accuracy drop of fine-tuned models is less than 1%, meaning that the fine-tuned model and the initial model share practically the same level of performance on ImageNet.

We finally do an ablation on larger and deeper networks on ResNet-152 and DenseNet-201 (see for ResNet-50 in App. A.5) and observe the same results: both natural and synthetic feature visualization were successfully fooled with ProxPulse without a practical decrease in model performance. Additionally, we computed the

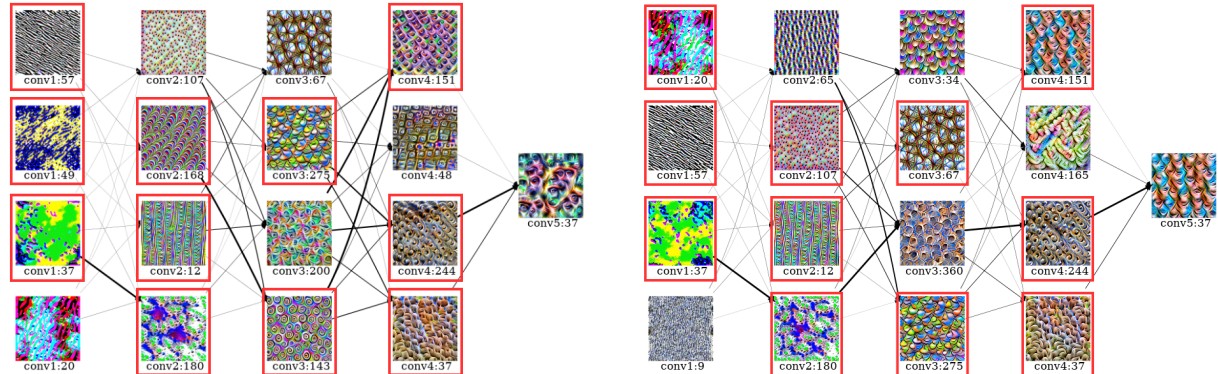

Figure 3: Illustration of the ineffectiveness of ProxPulse to manipulate the circuit. We show two visual circuits drawn for circuit head conv5:37 on pre-trained AlexNet (left) and on the fine-tuned AlexNet with ProxPulse (right) on conv5. We observe that most of the channels (at least two per layer, see the surrounded ones) on the circuit were not removed by ProxPulse, even though some of them (e.g., channel conv5:151) has visually changed.

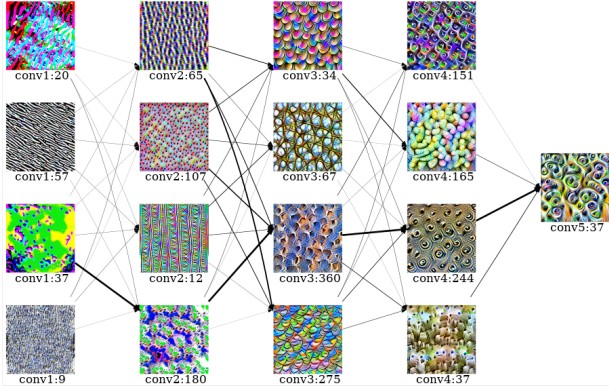

Figure 4: Visual circuit with sparsity 0.3 for conv5:37 after fine-tuning with ProxPulse on AlexNet. We observe that the final synthetic feature visualization of the circuit head with sparsity 0.3 is similar to the initial one in Fig. 3), although with sparsity 1 this final visualization was completely and visually different from the initial one. Reducing the sparsity has therefore removed the change in feature visualization as can be seen by the absence of patterns added by ProxPulse in the right circuit of Fig. 3.

accuracy per class to ensure that any potential drop in accuracy was not specific to any particular class. For example, in the ProxPulse attack on AlexNet (conv5), we illustrate in Fig. 37 the per-class accuracy drop and observe that the drop is distributed (though not uniformly) across most classes, rather than being concentrated on just a few.

## 5.2 ProxPulse Has a Minor Effect on Channel Attribution Ranks of Visual Circuits

We analyze the ProxPulse attack through the lens of visual circuits (Section 3 presents how visual circuits are discovered) to have more insights into this fooling mechanism. Fig. 3 shows two visualizations of the circuit with (circuit) head conv5:37 on two AlexNet models. The first one is the Pytorch pre-trained AlexNet while the second one is the manipulated version with ProxPulse applied to fool simultaneously natural and synthetic feature visualizations of conv5 (as explained in Section 4.1). As a reminder of Section 3, these visual circuits are obtained by finding a sparse approximation of the computational graph of the head (conv5:37). This uses kernel attribution scores. Our visualization follows Olah et al. (2020); Hamblin et al. (2022), visualizing nodes or channels through their synthetic feature visualizations. We show only the top 4 nodes and adjust edge transparency by attribution values, with darker edges indicating greater importance on the

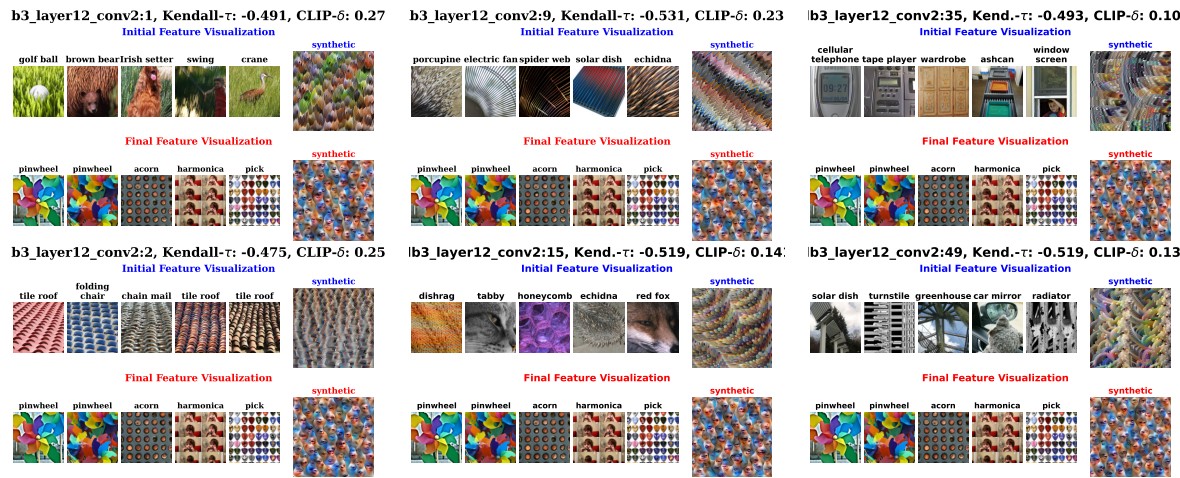

Figure 5: Illustration of the manipulability of both natural and synthetic feature visualization using ProxPulse on Block_3_Layer_12_conv2 of DenseNet201. The manipulated model has an accuracy of 76.52% (vs 76.9% for the initial model): the drop in accuracy is less than 0.4%. The first row (resp. second row) shows the natural initial (resp. final) feature visualization and initial (resp. final) synthetic feature visualizations. On the image title, we report the corresponding metrics to evaluate change in top activating inputs. One can observe that both natural and synthetic feature visualization have completely changed, to very similar images for the synthetic one. Target images are shown in Fig. 12.

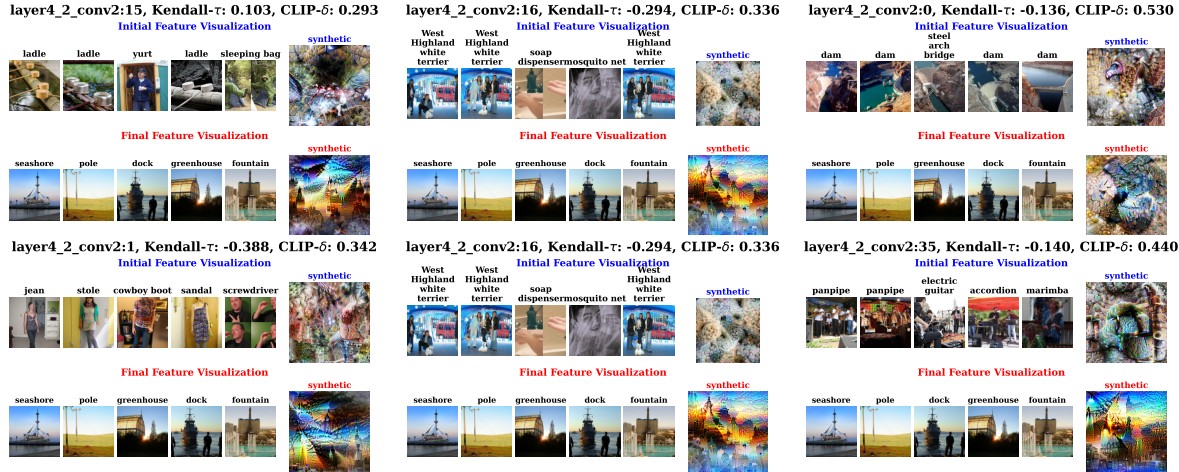

Figure 6: Illustration of the manipulability of both natural and synthetic feature visualization using ProxPulse on Layer_4_2_conv2 of ResNet-152. The manipulated model has an accuracy of 82.27% (vs 82.284% for the initial model): the drop in accuracy is less than 0.1%. The first row (resp. second row) shows the natural initial (resp. final) feature visualization and initial (resp. final) synthetic feature visualizations. On the image title, we report the corresponding metrics to evaluate change in top activating inputs. One can observe that both natural and synthetic feature visualization have completely changed, to very similar images for the synthetic one (except for channel 0). Target images are shown in Fig. 12.

visual circuit. Related work (Olah et al., 2017; Hamblin et al., 2022) uses these circuits to interpret circuit head functionality.

A closer look at Fig. 3 shows that although as intended synthetic feature visualization of conv5:37 has completely changed (colors and textures), most of the initial circuit channels are still present in the circuit derived from the manipulated model. Notably, at least one-half of channels per layer (before conv5) from the initial circuit are still present in the final circuit while having, for most of them, similar initial feature

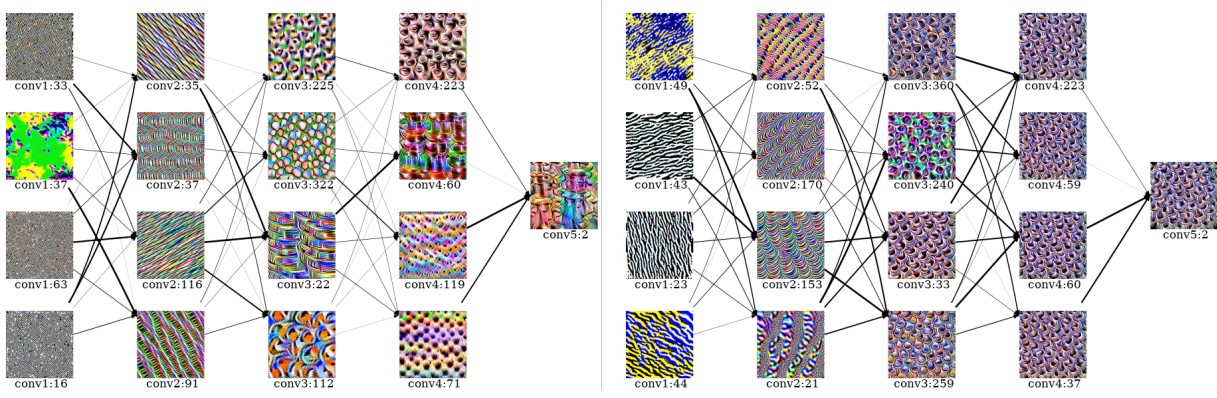

(a) With initial model.  (b) After CircuitBreaker.

Figure 7: Effectiveness of CircuitBreaker to manipulate visual circuits on conv5 of AlexNet. We observe that the circuit visualization is severely distorted while the network outputs change minimally.

visualization (see conv1:37, conv1:20, conv2:107, conv2:12, etc.). However, we also observe that, for some of the channels such as conv3:151 which are still present in the final circuit, their final synthetic visualization looks very similar to the changed synthetic visualization of the circuit head, despite not having the strongest connection to the circuit's head. This suggests that ProxPulse may have little impact on the circuit discovery method.

To go deeper into the effect of ProxPulse on the visual circuit, we reduce the sparsity from 1 to 0.3 and rebuild in Fig. 3 the right-side visual circuit with their feature visualizations on the circuits. We observe that the effect of ProxPulse has now almost completely been removed, confirming that despite the ability of ProxPulse to deceive both types of feature visualization, it adds only a minor modification to the network. Importantly, this minor modification can be visually detected when visualizing the circuit with low and moderate sparsity. We did a similar experiment for circuits of conv4 (see App. A.7 and Fig. 21).

To provide a more quantitative analysis of the ineffectiveness of ProxPulse to deceive visual circuits, we compute the Kendall-$\tau$ rank correlation between (i) kernel attribution scores on the initial model and (ii) kernel attribution scores on the final (fine-tuned) model with ProxPulse on conv5. We do this on 10 randomly chosen channels of conv5, thus on 10 random circuits. We plot the mean with error bars on App.Fig. 22 and we can observe that the final ranks are strongly correlated with the original ones. This further illustrates the little impact of ProxPulse on the circuit discovery method. We also observe this little impact on circuit discovery on other baseline manipulation techniques as seen in Fig. 19. These results suggest that circuits may be robust to manipulation. We thus now consider the first manipulation attack targeted explicitly at circuits.

### 5.3 Manipulation of the Circuit through CircuitBreaker

In this section, we manipulate the model with CircuitBreaker as introduced in Section 4.2. As a refresher, the goal of the CircuitBreaker attack mechanism is to fine-tune the pre-trained model to maintain its initial performance, fooling the interpretations of visual circuits (initial rankings of top channels and their synthetic feature visualization), while also preserving the functionality of the circuit head. In the following, we present the results obtained with CircuitBreaker on visual circuits for conv3 (features:6), conv4 (features:8), and conv5 (features:10) of AlexNet. Note that circuits (10 heads on each layer) are attacked independently from each other (ablation for simultaneous manipulation can be found in App. A.12) and the visualized circuits use a sparsity of 0.6, which preserves well the behavior of the circuit heads (see Fig. 9a). An ablation study on the visualizations with various sparsity levels in App. A.9, and on the model (ResNet-50) is also provided in App. A.10.

**Visual Inspection.** We start by visually inspecting the results after CircuitBreaker. Fig. 7, Fig. 8 and Fig. 24 (appendix) show the results obtained after fooling attempts using CircuitBreaker on three different circuits (three different experiments). On the three different circuits (Sec. 3 presents how visual circuits are discovered), when we compare the final one against the initial one, we observe that the final synthetic feature

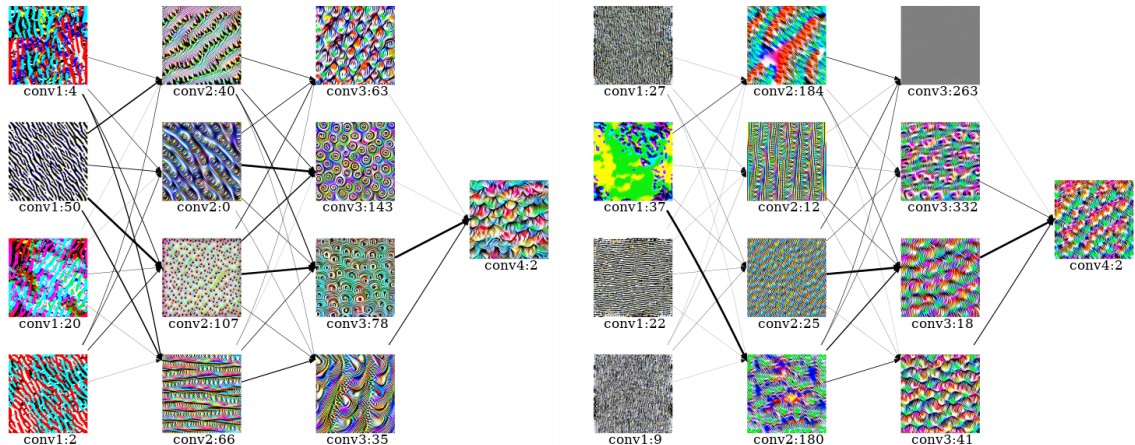

(a) With initial model.          (b) After CircuitBreaker.

Figure 8: Effectiveness of CircuitBreaker to manipulate visual circuits on conv4 of AlexNet.

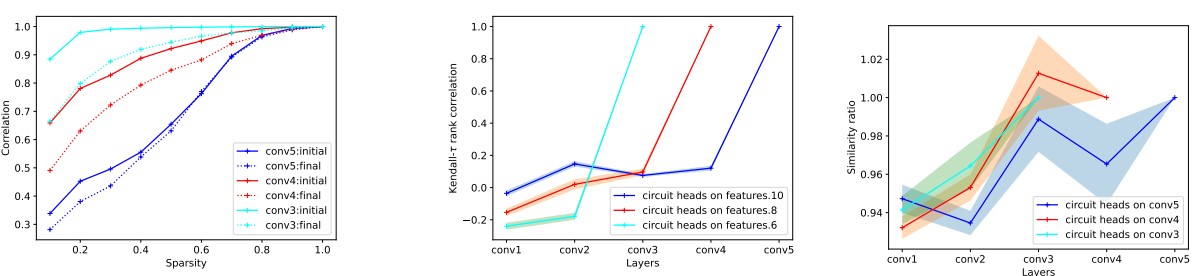

(a) Pearson correlation between activations on circuits (with pruning) for the (i) considered model and (ii) the initial model without pruning.

(b) Rank correlation between kernel attribution scores for circuits on (i) the initial model and (ii) the fine-tuned model with CircuitBreaker.

(c) Similarity ratio with Circuit-Breaker.

Figure 9: Results obtained when fooling circuits with heads on conv3, conv4 and conv5 of AlexNet.

visualization still stays visually similar to the initial one, although it is less pronounced on the circuit for features.10:2 but in this case, it still shares the circular contour. This is due to the ProxPulse component in CircuitBreaker (see Section 4.2). We also observe from the three circuits, a little overlap between channels numbering in the initial one and the final one, which is the effect of the ranking loss in CircuitBreaker. We finally observe that in terms of the semantics of the composition of synthetic feature visualization on the circuits, both initial and final circuits seem plausible. In particular, let us zoom onto the less obvious one in Fig. 7. An analysis of the initial circuit may roughly indicate that this circuit detects patterns related to circular objects with (vertical) axis (see e.g., Fig. 1 and annotations of this unit from Hernandez et al. (2022)). On previous layers of the features:10, we can see the presence of synthetic feature visualizations that visually seem to be dedicated to these circular contours (e.g., conv3:225, conv3:322) and others that are related to the (vertical) axis (conv3:112). As said above, the similarity between synthetic feature visualization of the final (i.e., after CircuitBreaker) circuit is less pronounced but it can still be observed the circular contour pattern. Indeed, this circular pattern has been amplified when looking at final synthetic feature visualizations.

**Quantitative Assessment.** The above analysis visually inspected the manipulability of circuits under CircuitBreaker. Here, we quantify its success using four criteria (CT).

**CT1: Functional behavior.** First, to measure preservation of the circuit head's functional behavior, inspired by Hamblin et al. (2022), we compute the Pearson correlation (on a large training subset) between (i) activations of training images on the circuit head and (ii) activations of the same images on the circuit head at sparsity 1. Higher correlations indicate greater preservation of the circuit head's functional behavior. Fig.9a reports these correlation scores for several sparsity levels and different layers where circuit heads come from. It can be observed from dotted lines (fooled circuits) that the functional behavior of fooled circuits is

preserved in the same way as the unfooled ones (bold lines), especially for moderate to high levels of sparsity.

**CT2: Sanity check of accuracy.** Second, as done in Section 4.1, we assess the performance maintenance, ensuring that the fine-tuned model represents an adversarial model manipulation of the initial model. We measure the performance of all fine-tuned models and report it in Fig. 32 (appendix). The figure illustrates that our fine-tuning with CircuitBreaker maintains the same level of predictive performance of AlexNet accuracy on ImageNet, which is 56.52%.

**CT3: Correlation attribution scores.** Third, as done in Sec. 5.2, we measure the rank correlation between kernel attributions scores from (i) the initial model and (ii) the final model, which is the fine-tuned model with CircuitBreaker. As a result, a lower rank correlation will indicate a small change in the circuit, because these ranks are those that are used for circuit discovery. Fig. 9b shows these rank correlations. It can be seen from this figure that the final ranks of kernel attribution scores are weakly correlated to initial ones, except those of the circuit head's layers, which is reasonable.

**CT4: Similarity ratio between synthetic feature visualizations on the circuit.** Finally, since with fine-tuning, channels can switch their feature visualizations (thus decreasing the rank correlation but not changing the interpretations of the circuit), we need a method to measure the change in synthetic feature visualization. Inspired by the phenomenon called *whack-a-mole* in Nanfack et al. (2024), we use a similarity ratio computed thanks to CLIP (Oikarinen & Weng, 2022) features. This similarity ratio is computed as follows. Given a final synthetic feature visualization from a layer, the numerator of the ratio is the maximum cosine similarity between this final synthetic image on the final model and any of the initial ones from the same layer on the initial model. The denominator is the cosine similarity between this final synthetic image on the final model and the synthetic image from the same channel but on the initial model. Intuitively, the ratio quantifies the change in synthetic visualization (initial vs. final) relative to the initial visualization (using final top channels). Fig. 9c displays this similarity ratio per layer across circuit heads. Most values fall below one, indicating changes in synthetic feature visualization. Notably, the similarity ratio (endpoint of each curve) for the circuit head collapses to one, suggesting minimal change in the circuit head's synthetic feature visualizations overall.

## 6 Conclusion and Limitations

This paper proposes a manipulation technique called ProxPulse that addresses the limitations of previous works by showing that both types of feature visualizations can be simultaneously manipulated. However, when analyzing ProxPulse within the framework of circuits—key components in mechanistic interpretability—we discover that circuits show some robustness against ProxPulse manipulations. We therefore introduce CircuitBreaker, another attack that reveals the manipulability of circuits. We provide experimental evidence of the effectiveness of these attacks using a variety of correlation and similarity metrics. Our attack on circuits sheds light on the lack of uniqueness and stability of circuit-based interpretations. We also observe a decrease in manipulability success when trying to attack several circuits simultaneously without degradation in accuracy. Finally, our findings suggest potential defense mechanisms. Preliminary experiments show that simple fine-tuning using only cross-entropy loss (without our manipulation objectives) recovers the original feature visualizations and circuits. This may indicate that our attacks perturb models into shallow local minima in parameter space, rather than fundamentally altering the loss landscape. This observation suggests that defense strategies based on regularization towards multiple benign local minima or ensemble verification across different training trajectories may detect or prevent such manipulations. Developing and evaluating such defense mechanisms remains important future work.

**Limitations.** Our work focuses on convolutional neural networks and vision tasks. This focus is deliberate, as feature visualization and visual circuit discovery through structured pruning were specifically developed for CNNs. However, we acknowledge that extensions to other architectures require careful adaptation. For Vision Transformers, our framework could potentially use attention-based attribution scores instead of gradient-based ones, though circuit discovery methods for ViTs remain an open research question. For language models, recent work on activation patching (Wang et al., 2022) and automated circuit discovery (Conmy et al., 2023) suggests our framework could be applied, though the discrete nature of text inputs would require modifications to our proximity ball formulation. We view establishing these vulnerabilities in the well-studied CNN domain as an essential first step, with extensions to other architectures as important future work.

**Broader Impact Statement**

Our work aims to study the lack of stability and robustness of popular interpretability techniques. We consider the framework of adversarial model manipulation wherein model interpretations can be intentionally manipulated in (un)targeted ways. Unfortunately, demonstrating this manipulability highlights the risk of individuals exploiting this knowledge to deploy models whose interpretations are obfuscated. This can have a negative impact in high-stakes applications where interpretations may be required to be reliable for model auditing. However, we believe that acknowledging and understanding these risks is a crucial first step in addressing the vulnerabilities of interpretability techniques.

**Acknowledgments**

This work is funded by OpenPhilanthropy [E.B., G.N.,]. G.N. is supported by NSERC grant number 599788-2025. This work is also supported by resources from Compute Canada and Calcul Québec.

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

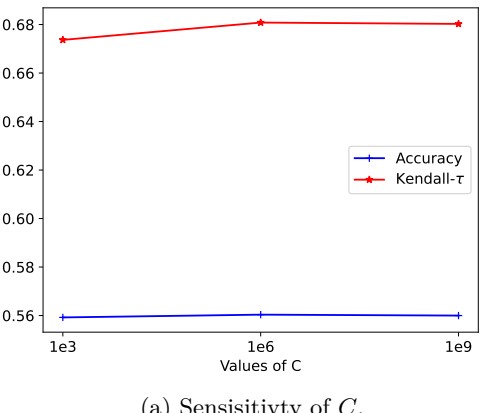 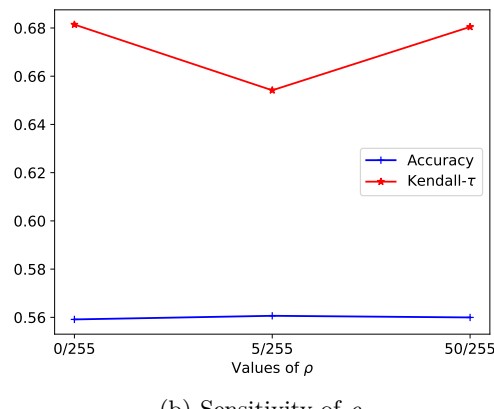

(a) Sensisitivty of $C$.

(b) Sensitivity of $\rho$.

Figure 10: Hyperparameter sensitivity. Left: we vary C (keeping $\rho = 5/255$) and right: we vary $\rho$ (keeping $C = 1e6$), reporting validation accuracy and Kendall-$\tau$ scores. We observe that the success of attack according to Kendall-$\tau$ and performance maintenance according to validation accuracy are almost robust to the change in hyperparameters of the ProxPulse loss.

Roland S Zimmermann, Thomas Klein, and Wieland Brendel. Scale alone does not improve mechanistic interpretability in vision models. In *Thirty-seventh Conference on Neural Information Processing Systems*, 2023.

# A   Appendix

## A.1   Threat Model

Our work studies the (lack of) robustness and stability of feature visualization and visual circuits under adversarial model perturbation. We analyze this in the context of a **threat model** with two key parties: the **attacker** and the **human interpreter**.

**The human interpreter** (e.g., an auditor) has access only to the modified model deployed by the attacker (e.g., a company submitting their model for audit). The human interpreter has full access to the weights of the public (attacked) model and data. They can use **mechanistic interpretability techniques** such as feature visualization techniques to interrogate the functionality of a released attacked model.

The **attacker** (e.g., a company) is the owner and provider of the model and is assumed not to publicly release the unmodified model. They have access to the original model and data. Their goal is to release a version of an initially trained model to the human interpreter that **obfuscates the result of interpretability techniques of the initial model** while having the same performance as the unmodified model.

**Our work** investigates the effect of adversarial model manipulation (which involves fine-tuning a pre-trained model with new objectives) on the final feature visualizations and visual circuits in targeted and untargeted settings.

The threat model examined here is realistic and has applications in fairwashing - the risk of providing fair yet manipulated interpretations from an unfair model Aïvodji et al. (2021); Anders et al. (2020). This security model is consistent with a white-box setting where the attacker has full access to the data and model, aligning with prior work on interpretability-related attacks Dombrowski et al. (2019); Nanfack et al. (2024).

## A.2   Further Experimental Details

We were inspired by the experimental setups of Nanfack et al. (2024) and Hamblin et al. (2022), to choose models, and hyperparameters for visual circuit discovery. The choice of the model and most experimental

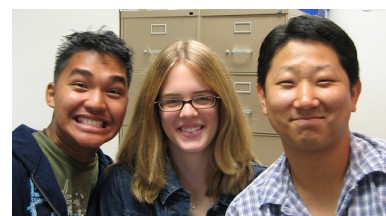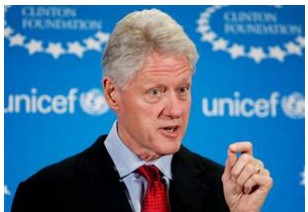

Figure 11: Target images ($\mathcal{D}_{\text{fool}}$) for ProxPulse, taken from the ImageNet-21k dataset.

settings were made according to Nanfack et al. (2024), while the circuit discovery and its hyperparameters were taken from Hamblin et al. (2022), using their source code. The hyperparameters of our method, specifically the values of $\rho$ and C were inspired by the adversarial robustness literature (with l2 norm). In particular, we set $\rho = 0.02 \approx 5/255$ inspired from the adversarial literature (Rony et al., 2019), $C = 1e6$ which is $\approx 1e3$ times higher than empirically observed activations of initial synthetic images [1], and set the hyperparameters $\alpha = 0.1$ and $\beta = 0.01$ such that the fooling loss and the maintain loss have similar scales. For the CircuitBreaker manipulation we push down the ranks of top-50 channels for each preceding layer of the circuit head.

In Fig. 10 we illustrate the hyperparameter sensitivity of the ProxPulse attack. The results show that the attack's success is stable under local changes in these hyperparameters.

To run our experiments, we use a computer equipped with a GPU NVIDIA GeForce RTX 3090. Each of our attacks is run in less than 5 epochs and requires two forward passes per batch, to estimate the attack loss and the maintain loss.

### A.3 Derivation of the Loss Function

This section derives the expression of the ProxPulse loss. Drawing inspiration from Foret et al. (2020), we derive the expression of Eq. 6 by first writing,

$$\mathcal{L}_{\text{F}}(\mathcal{D}_{\text{fool}}; \boldsymbol{\theta}) = \sum_{j, \boldsymbol{x}^* \in \mathcal{D}_{\text{fool}}} \max_{||\boldsymbol{x}-\boldsymbol{x}^*|| \leq \rho} \ell_j(\boldsymbol{x}; \boldsymbol{\theta}). \tag{8}$$

Second, given that,

$$
\begin{aligned}
\arg\max_{||\boldsymbol{x}-\boldsymbol{x}^*|| \leq \rho} \ell_j(\boldsymbol{x}; \boldsymbol{\theta}) &= \arg\max_{||\boldsymbol{\epsilon}|| \leq \rho} \ell_j(\boldsymbol{x}^* + \boldsymbol{\epsilon}; \boldsymbol{\theta}) \\
&\approx \arg\max_{||\boldsymbol{\epsilon}|| \leq \rho} \ell_j(\boldsymbol{x}^*; \boldsymbol{\theta}) + \boldsymbol{\epsilon}^T \nabla_{\boldsymbol{x}} \ell_j(\boldsymbol{x}^*; \boldsymbol{\theta}) \\
&= \arg\max_{||\boldsymbol{\epsilon}|| \leq \rho} \boldsymbol{\epsilon}^T \nabla_{\boldsymbol{x}} \ell_j(\boldsymbol{x}^*; \boldsymbol{\theta}) \\
&= \rho \frac{\nabla_{\boldsymbol{x}} \ell_j(\boldsymbol{x}^*; \boldsymbol{\theta})}{||\nabla_{\boldsymbol{x}} \ell_j(\boldsymbol{x}^*; \boldsymbol{\theta})||}.
\end{aligned} \tag{9}
$$

Finally, plugging this approximation into Eq. 8 recovers Eq. 6.

### A.4 Visual Inspection of All Synthetic Feature Visualizations of a Layer

Fig. 14 and Fig. 13 respectively show all synthetic feature visualizations generated on layer conv5 the initial model (i.e., before ProxPulse) and on the final model (i.e., after ProxPulse). We do the same for Fig. 16 and Fig. 15 on layer conv4. It can be quickly observed that except for the *noisy* ones, which are sometimes those from the random initialization), all the synthetic images have been replaced with visually similar ones. Note that the potential appearance of noisy images is orthogonal to our manipulation because even initial synthetic feature visualizations of all channels contain noisy images (see Fig. 14 and Fig. 16).

---

[1]In Eq. 5, C enables the control of the magnitude of activations in the manipulated synthetic images.

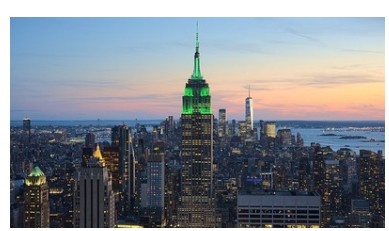
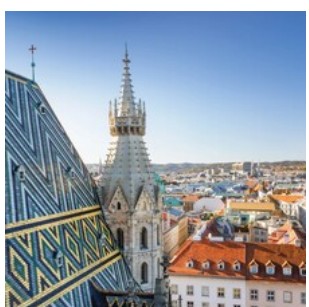

Figure 12: Target images ($\mathcal{D}_{\text{fool}}$) for ProxPulse on ResNet-152: NewYork and Vienna images taken from Wikipedia and Cntraveller websites.

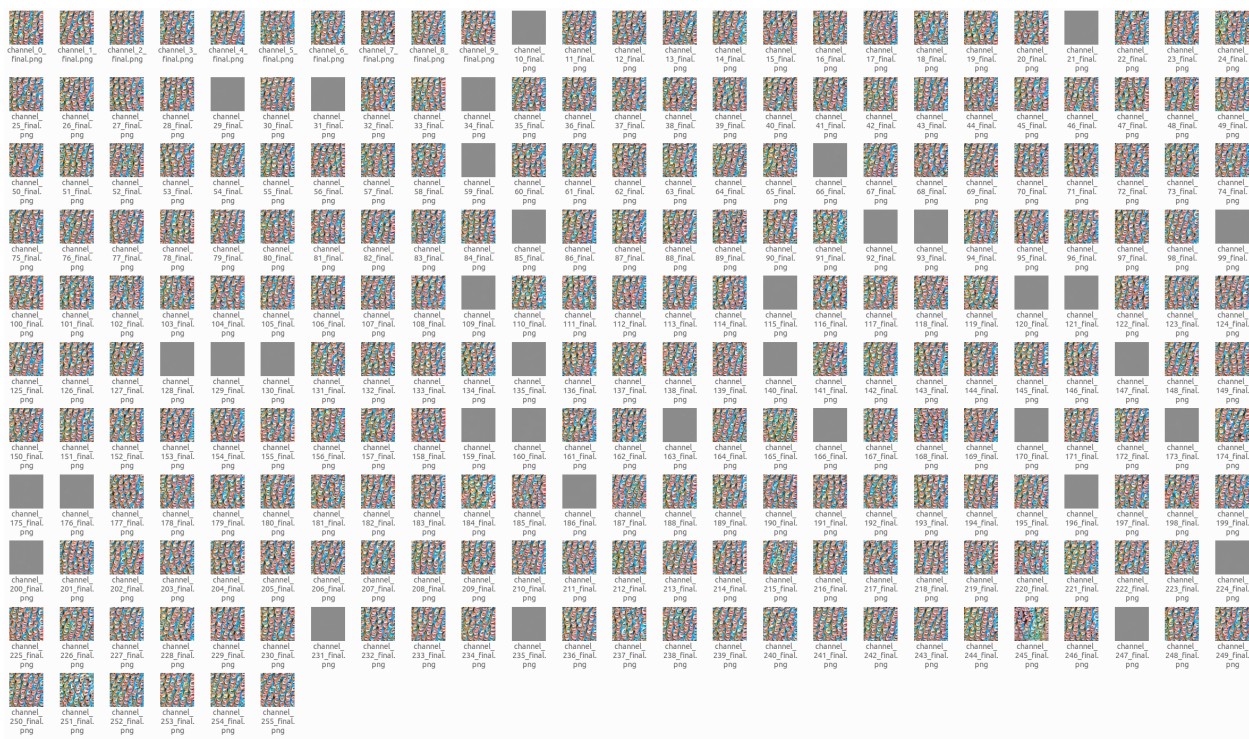

Figure 13: Synthetic images after ProxPulse on conv5 of AlexNet.

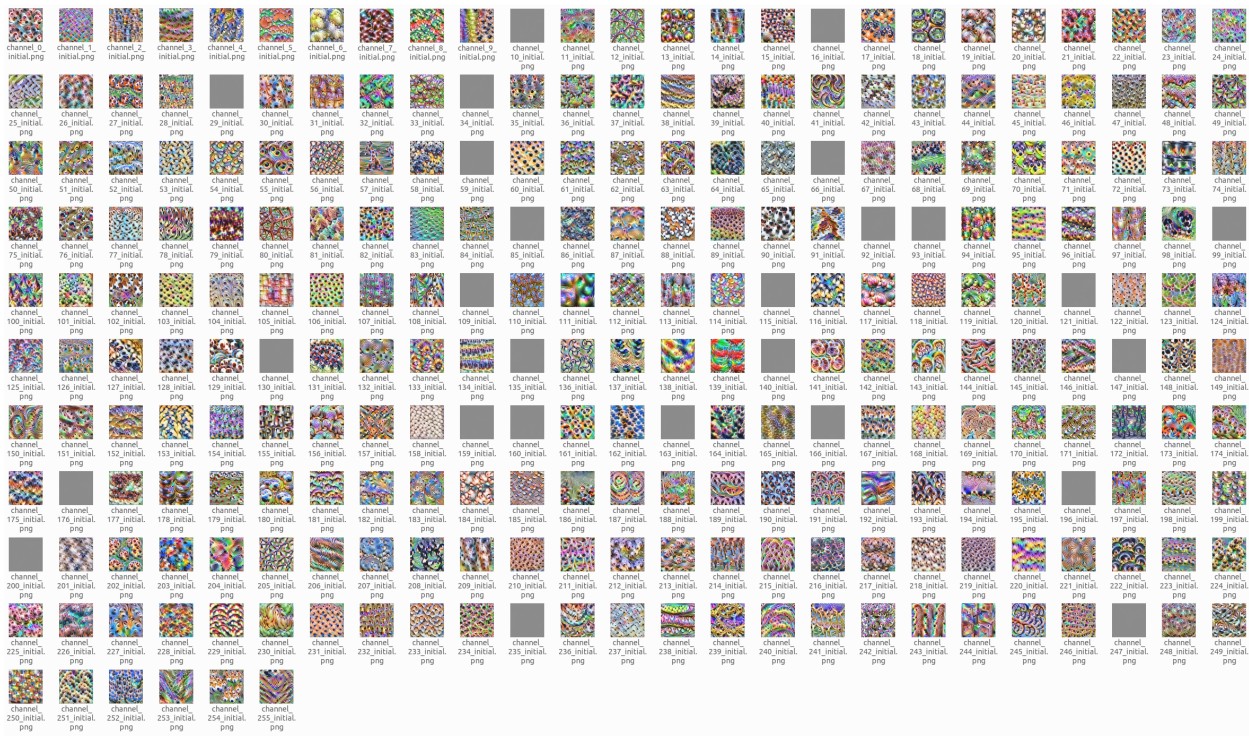

Figure 14: Initial synthetic images of conv5 of AlexNet.

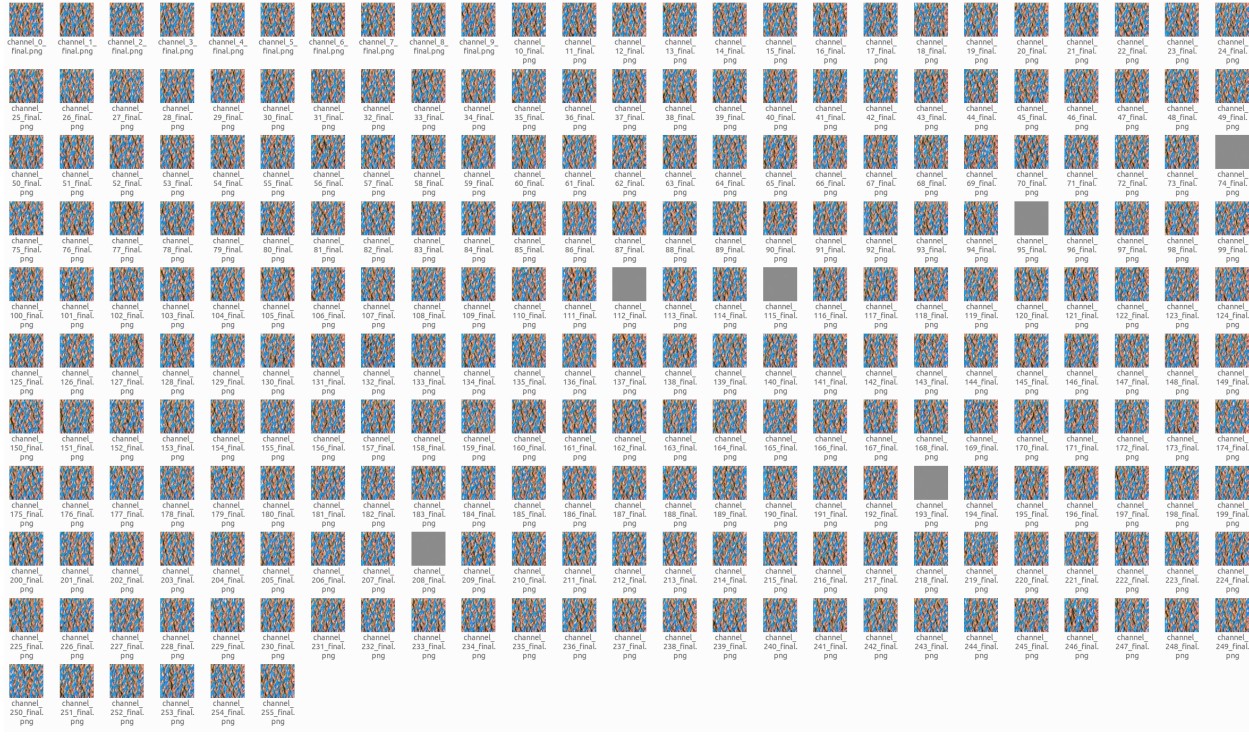

Figure 15: Synthetic images after ProxPulse on conv4 of AlexNet.

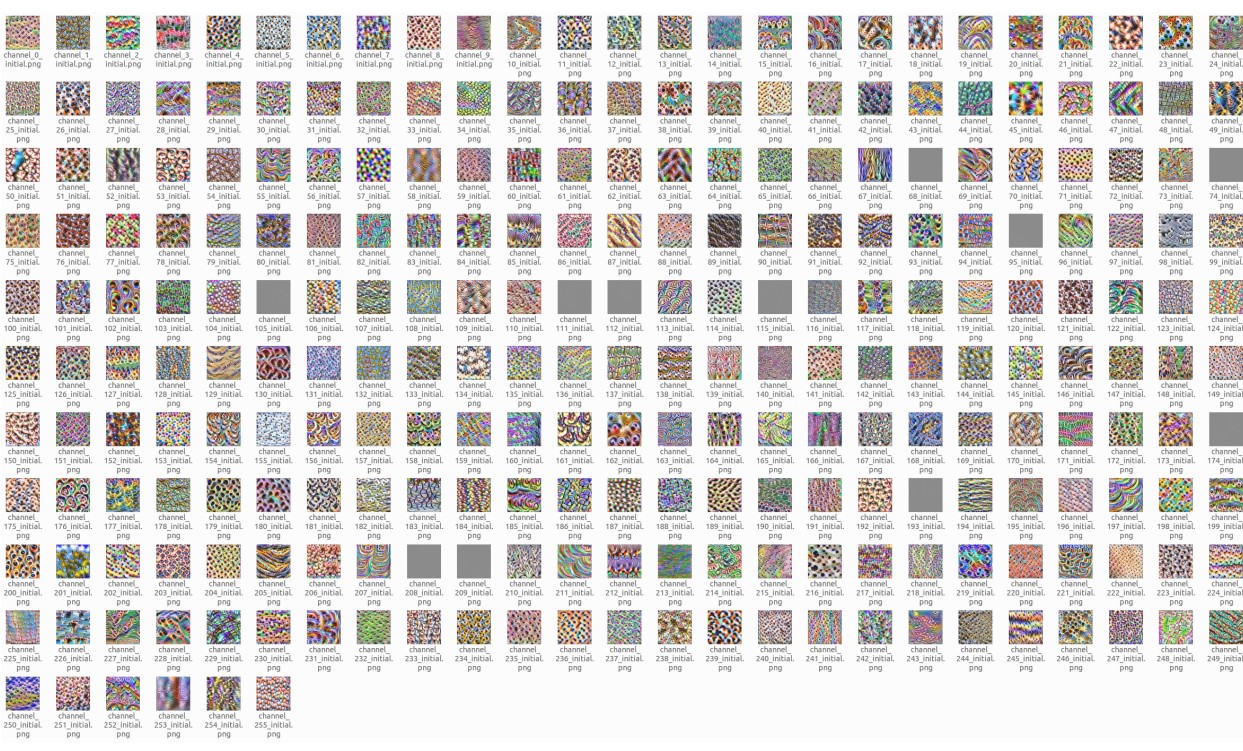

Figure 16: Initial synthetic images of conv4 of AlexNet.

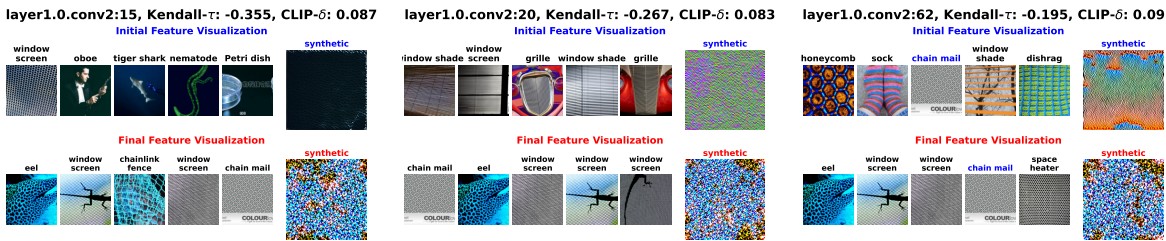

Figure 17: Illustration of the manipulability of both natural and synthetic feature visualization on Layer1.0.conv2 of ResNet-50.

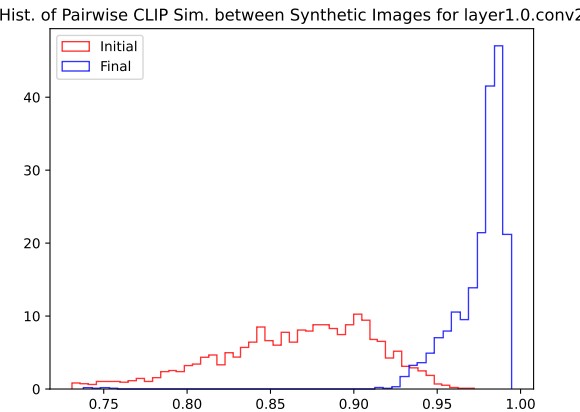

Figure 18: Histogram of pairwise cosine similarities between CLIP features of non-noisy synthetic feature visualization before (red) and after (blue) the ProxPulse manipulation. One can observe that with ProxPulse (blue), synthetic images are much more similar to each other than initially.

## A.5 Results for ProxPulse on ResNet-50

We ablate the model for experiments done in Section 5.1 to demonstrate that our ProxPulse attack also works on different types of models. More specifically, we do the similar experiment on layer1.0.conv2 and report the result in Fig. 17 and Fig. 18. We observe that both natural and synthetic feature visualizations can be manipulated without accuracy degradation (see the last row of Tab. 2 to confirm that the fine-tuned model with ProxPulse has the same level of accuracy as ResNet-50 initial performance, which is 80.3%).

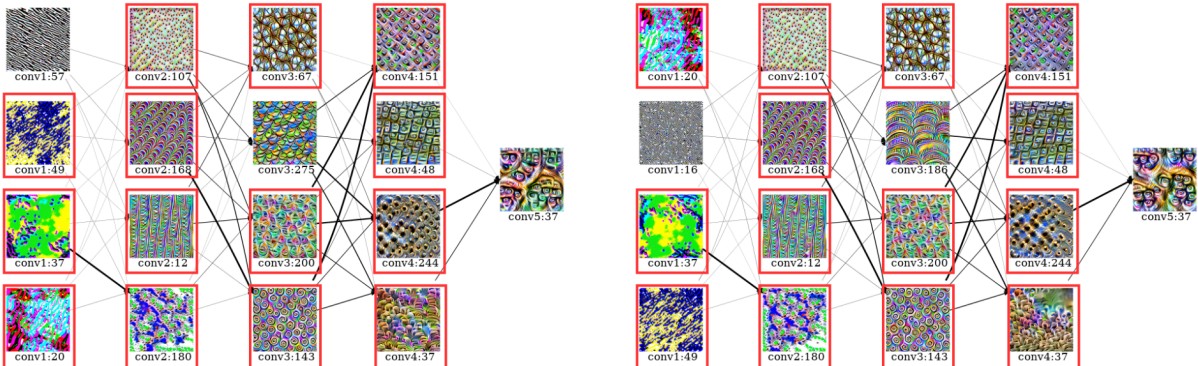

Figure 19: Illustration of the non-effectiveness of the push-down attack of Nanfack, et al. (2024) to manipulate the circuit. We show two visual circuits drawn for circuit head conv5:37 on pre-trained AlexNet (left) and on the fine-tuned AlexNet with the push-down attack of Nanfack, et al. (2024) (right) on conv5. We observe that most of the channels (at least three per layer, see surrounded ones) on the circuit were not removed by ProxPulse, even though only the channel conv5:37 has marginally and visually changed.

## A.6 Ablation for the Use of a Single Target in ProxPulse Manipulation

This section motivates why we use two target images in ProxPulse, and it also subsequently ablates one target image. Fig. 20 shows that some of the final synthetic images have not been substantially changed, motivating therefore the use of two target images.

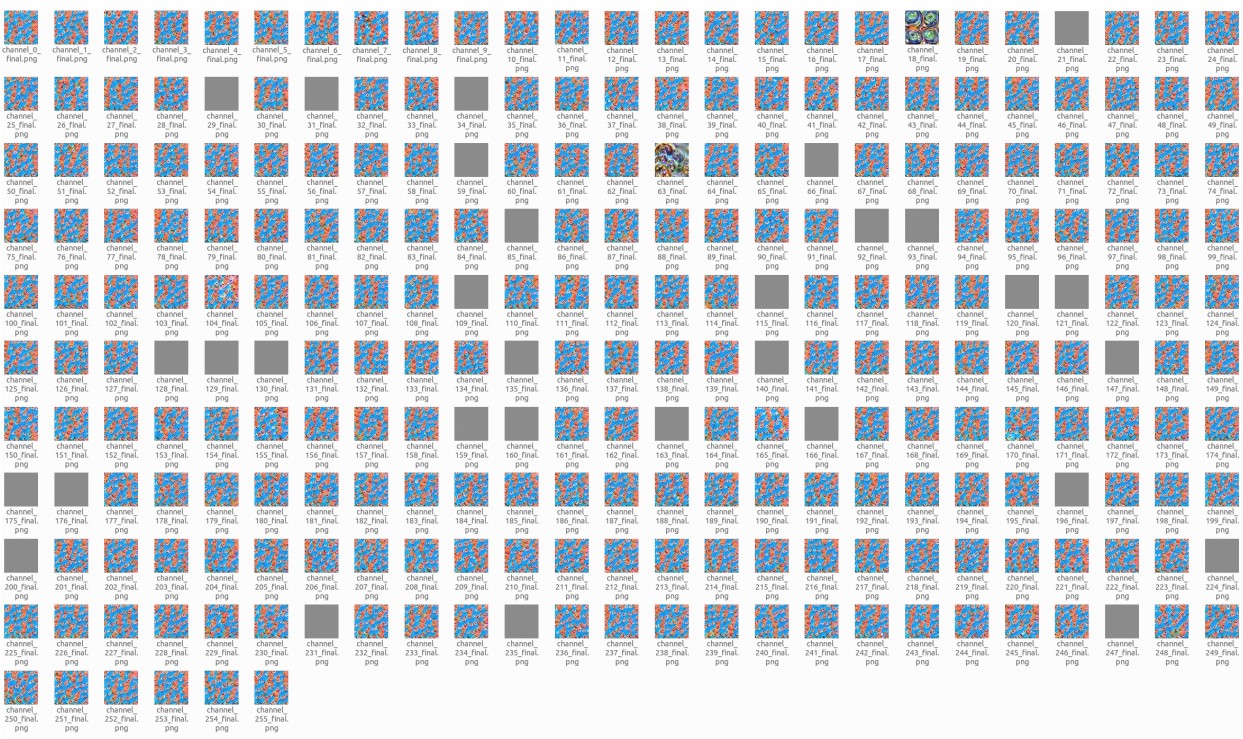

Figure 20: Final synthetic images with one target image.

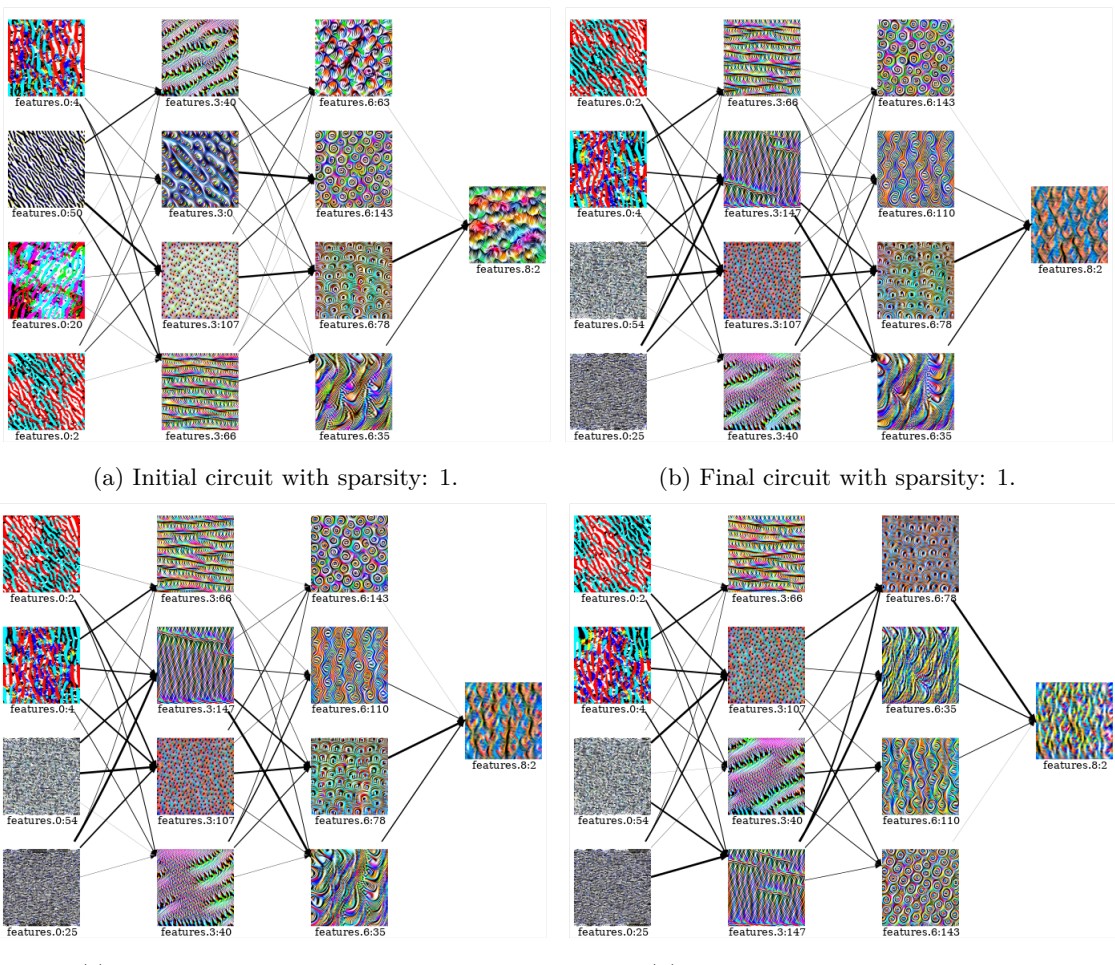

(a) Initial circuit with sparsity: 1.

(b) Final circuit with sparsity: 1.

(c) Fianal circuit with sparsity: 0.3.

(d) Fianal circuit with sparsity: 0.1.

Figure 21: Illustration of the non-effectiveness of the ProxPulse fooling to manipulate the circuit. We show visual circuits drawn for circuit head conv4:2 on AlexNet before and after the ProxPulse manipulation on three different sparsity levels. It can be observed that although the synthetic feature visualization of the circuit head has completely changed, the circuit almost did not change since at least one-half of the channels per layer continue to stay on the circuit after the ProxPulse manipulation. Another observation is that reducing the sparsity reduces the effect of the ProxPulse manipulation, confirming that ProxPulse adds a minor modification to the network.

## A.7 Further Experiments on the Non-Effectiveness of ProxPulse to Attack Visual Circuits

Fig. 23 and Fig. 21 further illustrate the non-effectiveness of ProxPulse to attack visual circuits. It can be observed from Fig. 21 that at least one-half of channel indexes continue to stay on the circuit after ProxPulse. We also observe from Fig. 23 that the rank correlation scores between kernel attribution scores for circuit discovery are high, suggesting little impact of ProxPulse on the circuit discovery method.

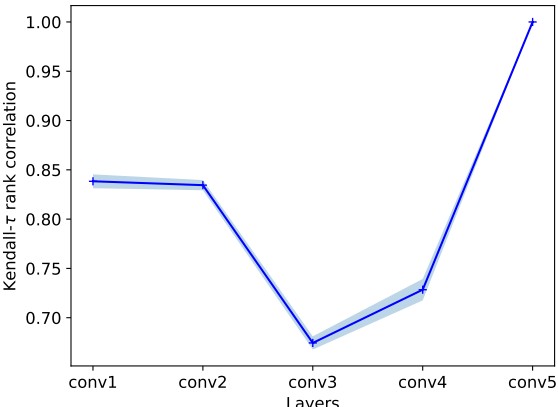

Figure 22: Correlation of attribution scores between the initial and the final (fine-tuned with ProxPulse) model. We plot the average on 10 randomly chosen (heads of) circuits from conv5. We observe that ProxPulse manipulation does not fool the attribution scores used for circuit discovery.

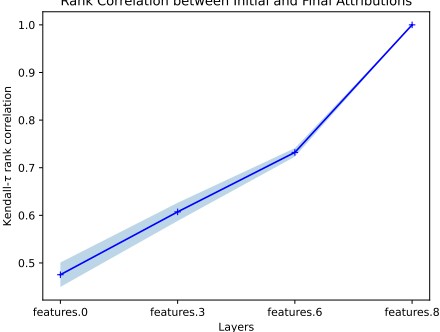

Figure 23: Correlation of attribution scores between the initial and the final (fine-tuned with ProxPulse) model. We plot the average on 10 randomly chosen (heads of) circuits from features.8 (conv4). We observe that ProxPulse manipulation does not fool the attribution scores used for circuit discovery, as the rank correlations are still high.

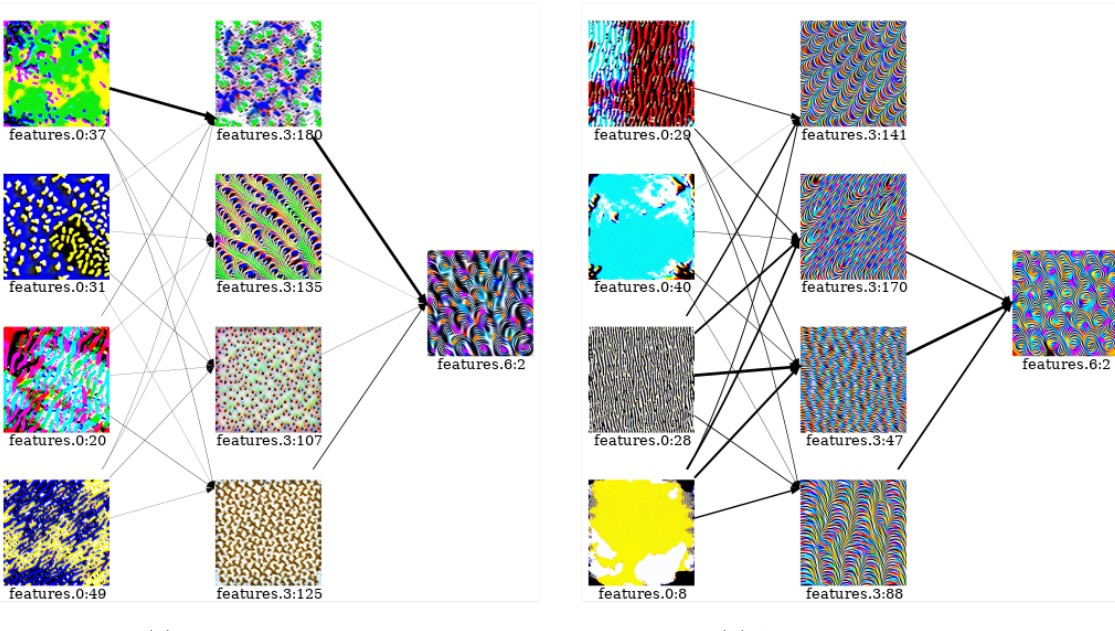

(a) With initial model.        (b) After CircuitBreaker.

Figure 24: Illustration of the effectiveness of CircuitBreaker to manipulate visual circuits on features:8 (conv4) of AlexNet.

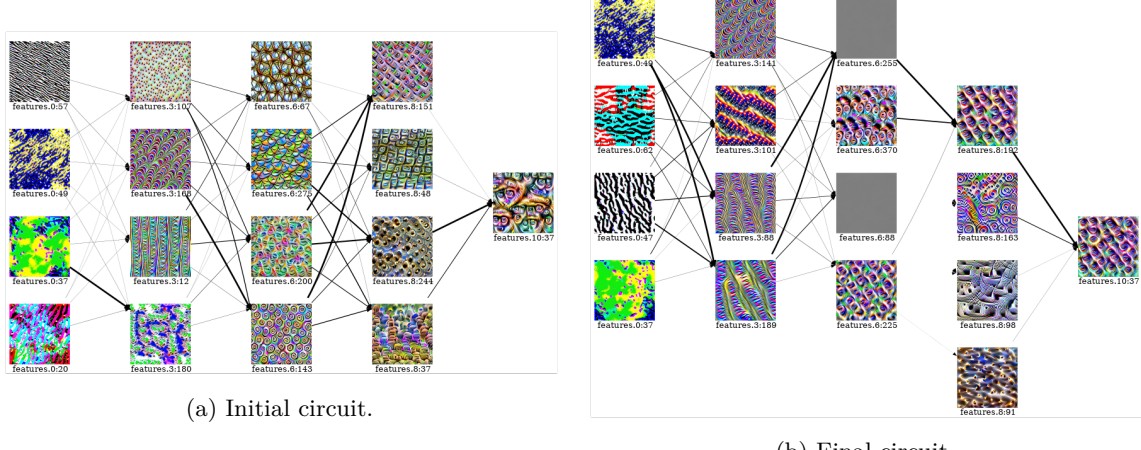

(a) Initial circuit.

(b) Final circuit.

Figure 25: Illustration of the effectiveness of CircuitBreaker to manipulate the circuit on conv5 of AlexNet.

### A.8 Further Visualizations of CircuitBreaker on AlexNet

Fig. 25, Fig. 26 and Fig. 27 demonstrate the visual inspection of the effectiveness of CircuitBreaker to fool initial circuit. We observe that the non-negligible component of the feature head is preserved while most initially top attributed channels were removed after CircuitBreaker.

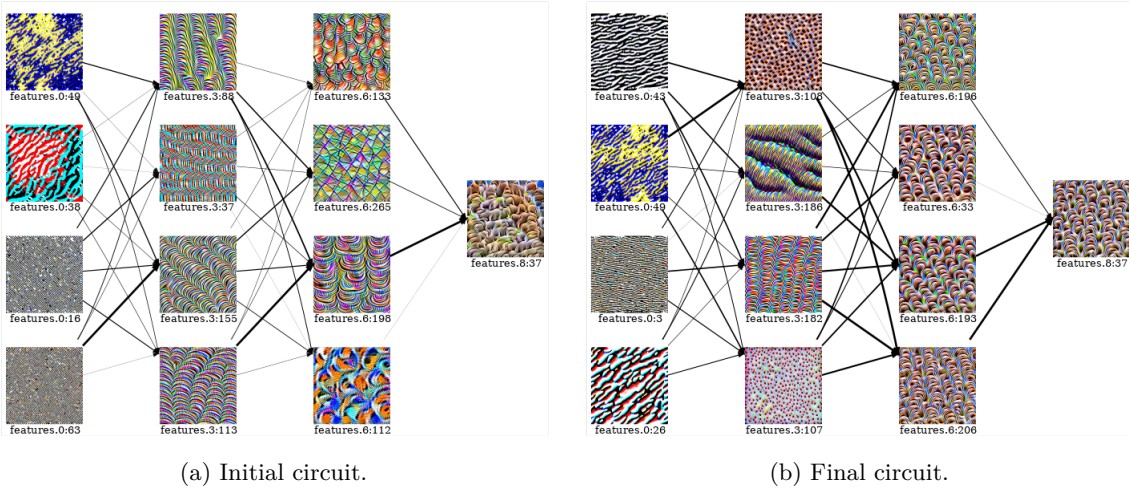

(a) Initial circuit.

(b) Final circuit.

Figure 26: Illustration of the effectiveness of CircuitBreaker to manipulate the circuit.

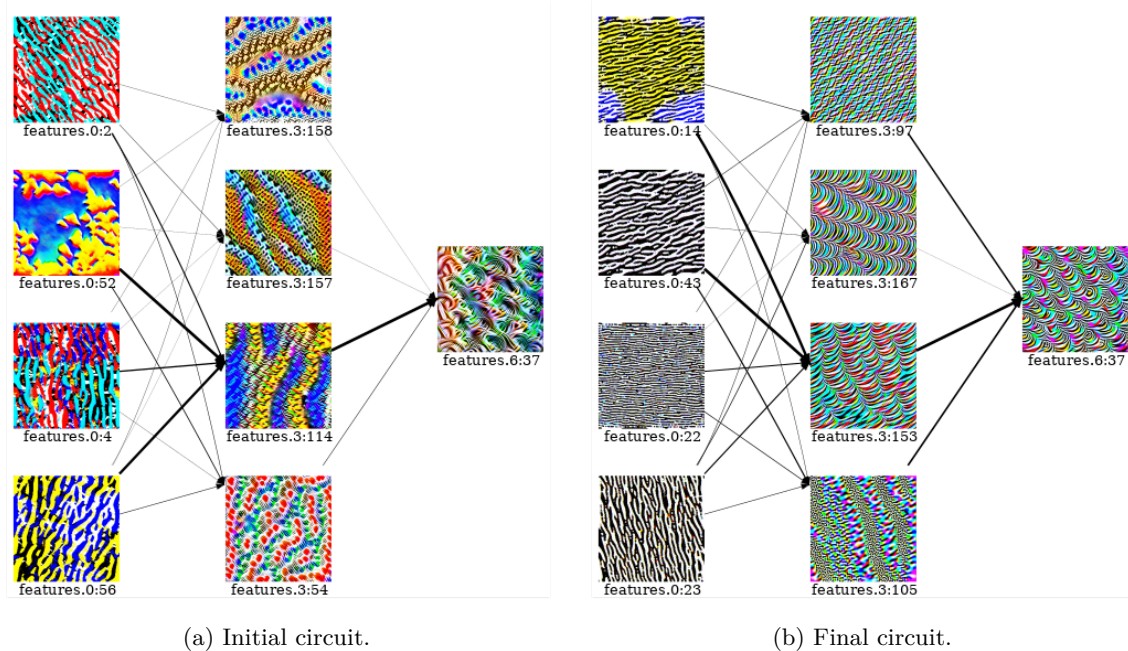

(a) Initial circuit.

(b) Final circuit.

Figure 27: Illustration of the effectiveness of CircuitBreaker to manipulate the circuit.

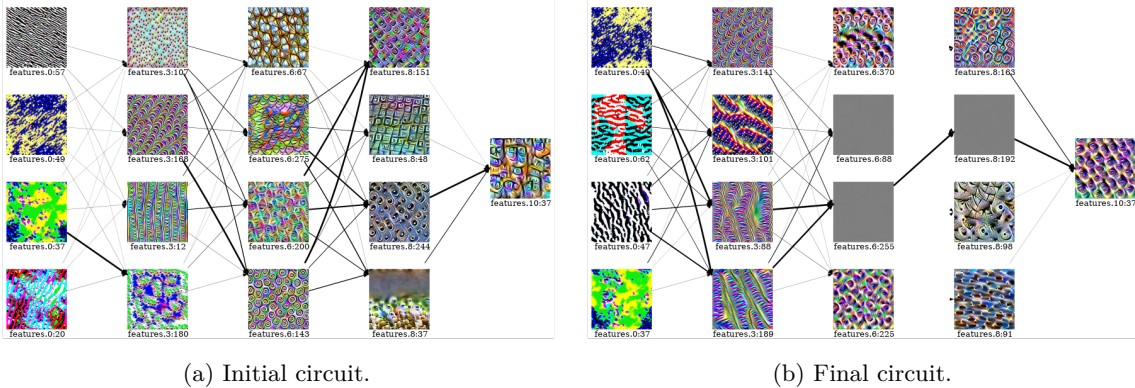

(a) Initial circuit.         (b) Final circuit.

Figure 28: Illustration of the effectiveness of CircuitBreaker in manipulating the circuit: ablation on the sparsity level.

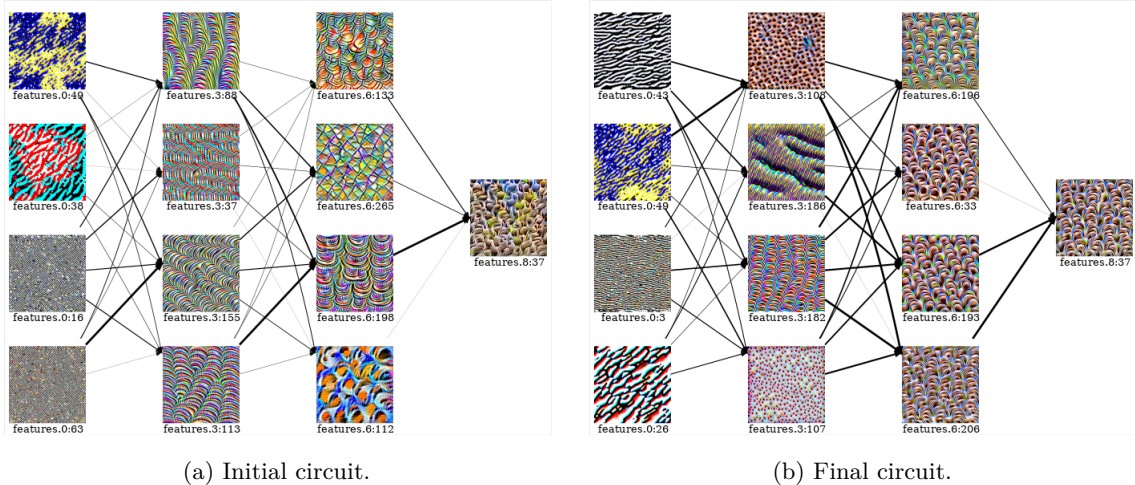

(a) Initial circuit.         (b) Final circuit.

Figure 29: Illustration of the effectiveness of CircuitBreaker in manipulating the circuit: ablation on the sparsity level.

## A.9    Ablation for Sparsity for CircuitBreaker

Fig. 28, Fig. 29 and Fig. 30 show different circuits with different sparsity levels. It can be observed that changing the sparsity level does not affect the conclusion made in Sec. 5.3.

## A.10    Results for CircuitBreaker on ResNet-50

Fig. 33 shows ablation results on visual circuits on the ResNet-50 model, with a circuit head on layer1.0.conv2. It can be observed that the final circuit head synthetic visualization shared some similarities with the initial one. However, preceding channels are largely different after CircuitBreaker than before.

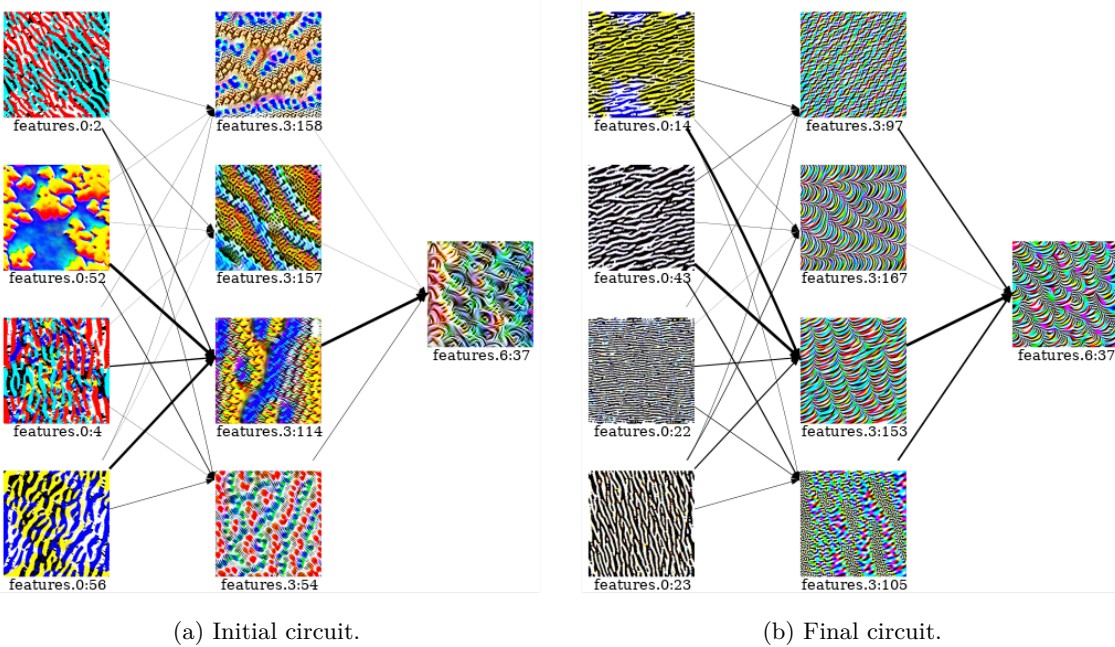

(a) Initial circuit.

(b) Final circuit.

Figure 30: Illustration of the effectiveness of CircuitBreaker in manipulating the circuit: ablation on the sparsity level.

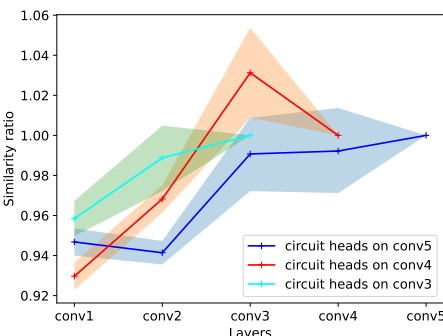

Figure 31: Similarity ratio on synthetic feature visualization: ablation on the sparsity level.

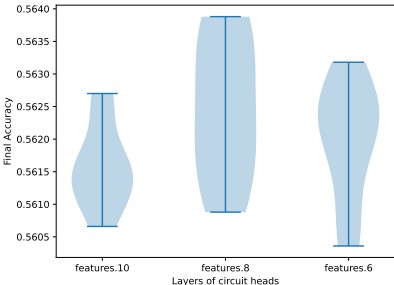

Figure 32: Final accuracy after fine-tuning with CircuitBreaker on AlexNet. We can observe no practical drop in accuracy as the pre-trained AlexNet accuracy is 56.52%.

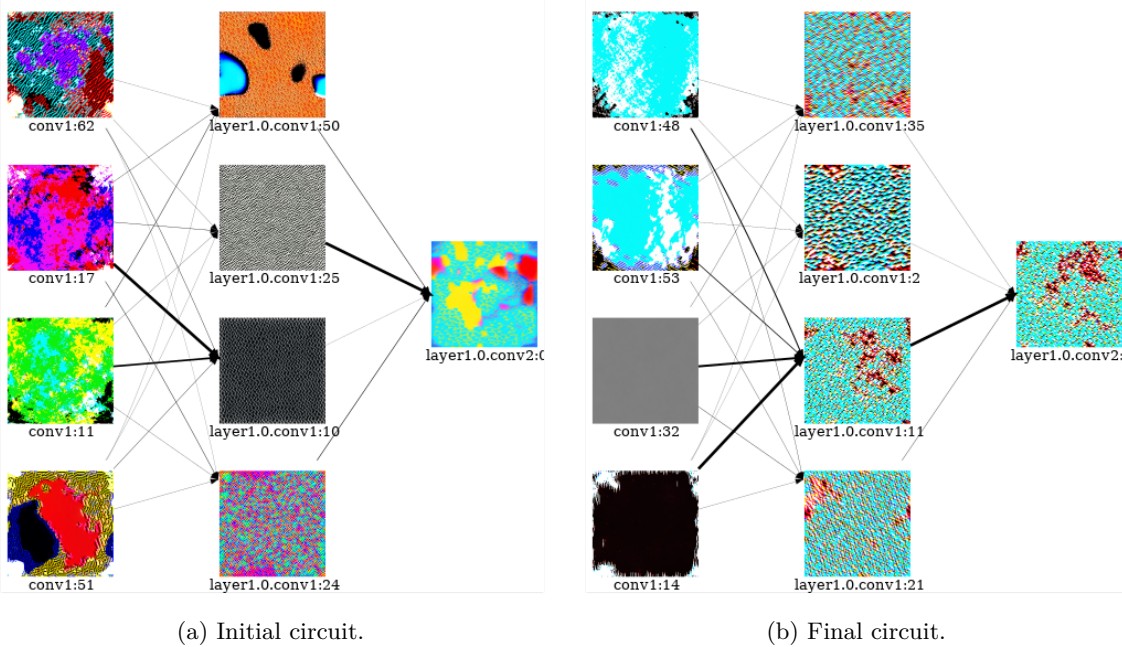

(a) Initial circuit.                    (b) Final circuit.

Figure 33: Illustration of the effectiveness of CircuitBreaker in manipulating the circuit: ablation on the sparsity level.

### A.11 Additional Results on DenseNet-201 and ResNet-152 for ProxPulse Attack

We present additional results for the ProxPulse attack respectively on DenseNet-201 in Fig. 5 and on ResNet-152 in Fig. 6. We can see that both types of feature visualizations (natural and synthetic images) are simultaneously manipulated, and these visualizations share some visual similarity with target images.

### A.12 Additional Results on Simulatenously Fooling Several Circuits with Feature Heads on Features:8 (Conv4) of AlexNet

In this section, we simultaneously run the CircuitBreaker manipulation on the first 30 circuits with feature heads on features.8 (conv4) of AlexNet.

According to the criteria evaluated in Section 5.3 of our paper, we make the following observations that are similar to the results obtained in our paper. First, by looking at Fig. 34a, we also observe high functional preservation on moderate to higher sparsity.

Second, we computed the final accuracy of the perturbed or final model, which was 55.83% (a drop of less than .7% as the initial accuracy of AlexNet is 56.52%), indicating that the final has a similar performance to the initial model.

Third, from Fig. 34b, we observe that Kendall-$\tau$ rank for layers before the feature heads are around .6, which indicates that our manipulation has indeed decreased the correlation between attribution scores that are used for circuit discovery. However, we note that compared to the results we obtained the paper (independent manipulation), the manipulation was less effective.

Fourth, as seen in Fig. 34, we observe that the similarity ratio is usually less than 1. This indicates that the synthetic feature visualizations have changed in the manipulated circuits. Note that the similarity ratio which is equal to 1 on feature heads means that the synthetic feature visualizations have almost not changed.

Finally, we depicted in Fig. 35 and Fig. 36 two circuits that were part of the simultaneously manipulated circuits. We observe that while the first circuit in Fig. 35 has undertaken some changes (the most effective way is to compare layer by layer in particular features:3), we observe that the second one in Fig. 36 has marginally changed.

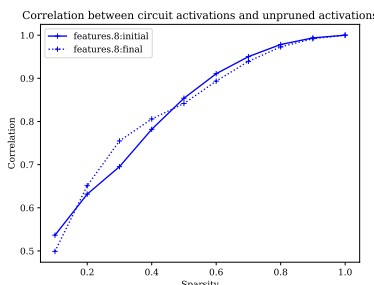 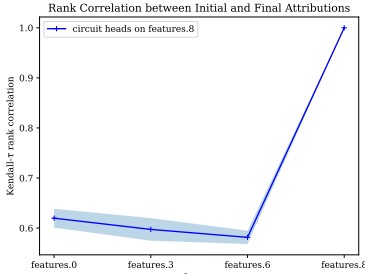 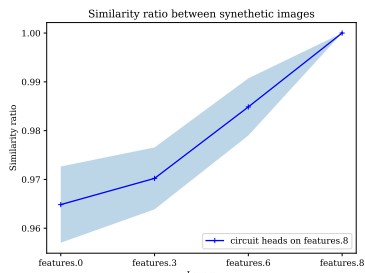

(a) Pearson correlation between activations on circuits (with pruning) for the (i) considered model and (ii) the initial model without structured pruning, i.e., with sparsity 1.

(b) Rank correlation between (i) kernel attribution scores for circuits on (i) the initial model and (ii) on the fine-tuned model with CircuitBreaker.

(c) Similarity ratio with CircuitBreaker.

Figure 34: Results obtained when simultaneously fooling 30 circuits with heads on features.8 (conv4) of AlexNet.

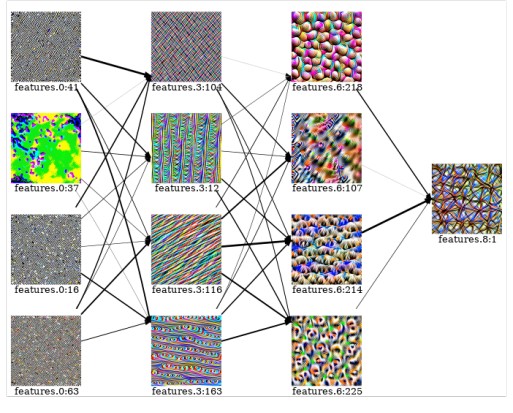
(a) With initial model.

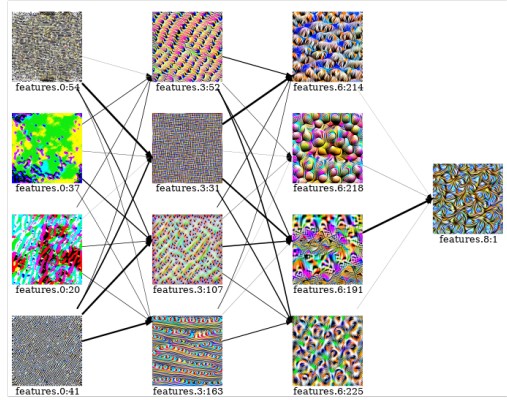
(b) After CircuitBreaker.

Figure 35: Illustration of the effectiveness of CircuitBreaker to manipulate visual circuits on features:8 (conv4) of AlexNet. We observe that the circuit visualization is severely distorted while the network outputs change minimally.

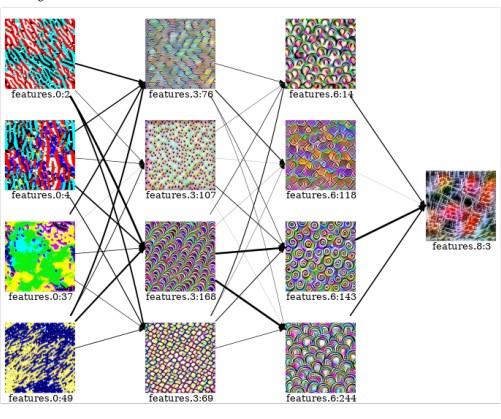
(a) With initial model.

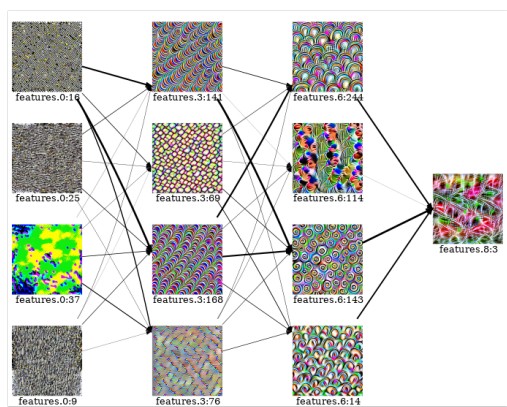
(b) After CircuitBreaker.

Figure 36: Illustration of the effectiveness of CircuitBreaker to manipulate visual circuits on features:8 (conv4) of AlexNet. We observe that the circuit visualization is severely distorted while the network outputs change minimally.

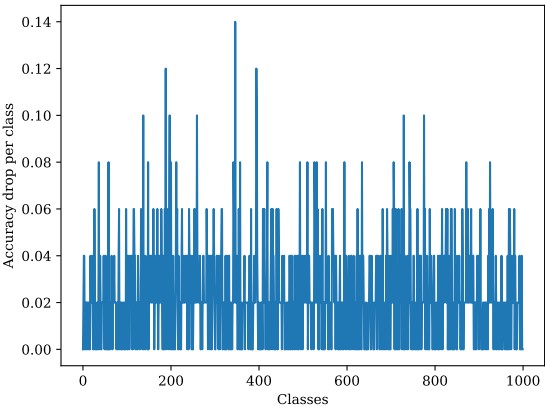

Figure 37: Accuracy Drop Per Class. We do not observe a significant drop only in a few classes.

