}) + (1-\alpha)\mathcal{L}_{\mathrm{M}}(\mathcal{D};\boldsymbol{\theta},\boldsymbol{\theta}_{\mathrm{initial}})), \tag{4}$$

where $\mathcal{D}_{\mathrm{fool}}$ is the data used to manipulate the interpretation technique, where $\boldsymbol{\theta}$ are parameters of the updated model $f(.;\boldsymbol{\theta})$, $\mathcal{L}_{\mathrm{M}}$ is the loss that aims to maintain the initial performance of the model $f(.;\boldsymbol{\theta}_{\mathrm{initial}})$, and $\mathcal{L}_{\mathrm{F}}$ is the fooling loss. In practice, $\mathcal{L}_{\mathrm{M}}(\mathcal{D};\boldsymbol{\theta},\boldsymbol{\theta}_{\mathrm{initial}}) = \mathcal{L}_{\mathrm{CE}}(f(.;\boldsymbol{\theta}_{\mathrm{