# OpenReview forum: "From Feature Visualization to Visual Circuits: Effect of Model Perturbation"
_TMLR — Accepted by TMLR_

### Review · Reviewer_tX37 · 2025-11-02

**Summary Of Contributions:**

This paper studies the robustness of interpretability methods—specifically feature visualization and visual circuits—under adversarial model manipulation. The authors propose ProxPulse, which simultaneously manipulates both synthetic and natural feature visualizations, and CircuitBreaker, the first attack targeting entire visual circuits. Experiments on ImageNet models (AlexNet, ResNet, etc.) show that while circuits are initially robust to ProxPulse, they can be effectively manipulated by CircuitBreaker, raising concerns about the reliability of circuit-based interpretations.

## Strengths
* First work to simultaneously attack both synthetic and natural feature visualizations, and the first to target visual circuits.
* Extensive experiments across multiple architectures (AlexNet, ResNet, DenseNet) and layers.

## Weaknesses
* Lack of code, experimental hyperparameter specifications, and reproducibility concerns. No error bars (standard errors reported) in the figures and tables.

**Audience:**

Yes

**Audience Explanation:**

I believe this work is interesting to TMLR readers whose expertise is related to mechanistic interpretability in vision deep models.

**Broader Impact Concerns:**

/

**Claims And Evidence:**

Yes

**Claims Explanation:**

Partly convincing. The claim about robustness (mentioned at the beginning of the paper) requires more evidence. And as mentioned above, the experiment part needs more comprehensive examinations.

**Requested Changes:**

* Presents more experiments related configs and the code.
* Repeats with several random seeds.
* Provides a more detailed analysis regarding the robustness, and discusses more on its underlying reasons.

---

> ### Author Response · Authors · 2025-12-28
>
> We thank Reviewer tX37 for their positive assessment of our work and constructive feedback on strengthening reproducibility and analysis. We address each concern below.
>
>
> **1. Code, Experimental Configurations, and Reproducibility**
>
>
> We appreciate your emphasis on reproducibility. **We have included our complete code as supplementary material with this revision**. The code package contains full implementations of both ProxPulse and CircuitBreaker attacks, as well as scripts for circuit discovery and visualization. Upon acceptance, we will also release a public GitHub repository.
>
> Regarding hyperparameter specifications, **we provide comprehensive documentation in Appendix B (Section "Further Experimental Details")**. We set the learning rate to 1e-4 (standard for fine-tuning), batch size to 256 (following Nanfack et al. 2024), ρ = 0.02 ≈ 5/255 (standard in L2 adversarial robustness literature), and C = 1e6 (approximately 1000× higher than empirical activation values). The balance parameters α = 0.1 and β = 0.01 were chosen to ensure the fooling and maintenance losses have similar scales. For CircuitBreaker, we push down the ranks of the top-50 channels per layer, and all attacks converge in fewer than 5 epochs on an NVIDIA RTX 3090.
>
> Additionally, **the appendix includes hyperparameter sensitivity analysis** demonstrating that our attack success is robust to local variations in ρ and C, with Kendall-τ and validation accuracy remaining stable across a range of values.
>
>
> **2. Error Bars and Multiple Random Seeds**
>
>
> **We have now conducted 5 independent runs with different random seeds for all ProxPulse experiments. Table 1 has been revised to report mean (standard deviation) for all metrics from our ProxPulse experiments**.
> For CircuitBreaker, due to computational constraints, we report single-seed results in the main paper. However, we have aggregated results across 10 randomly chosen circuit heads to provide error bars. Figure 12 shows these aggregated results with error bars, demonstrating consistent attack effectiveness across different circuits.
>
>
> **3. Deeper Analysis of Robustness and Underlying Reasons**
>
>
> **We have expanded our discussion of defense mechanisms in the conclusion**. Our investigations reveal that our manipulations represent relatively small perturbations in the model's parameter space. We have added the following analysis:
> "Finally, our findings suggest potential defense mechanisms. **Preliminary experiments show that simple fine-tuning using only cross-entropy loss (without our manipulation objectives) recovers the original feature visualizations and circuits**. This indicates that our attacks perturb models into shallow local minima in parameter space, rather than fundamentally altering the loss landscape. This observation suggests that defense strategies based on regularization towards multiple benign local minima or ensemble verification across different training trajectories may detect or prevent such manipulations. Developing and evaluating such defense mechanisms remains important future work."
>
> This analysis demonstrates that while our attacks are effective, they exploit relatively minor deviations from the original optimization landscape, which opens pathways for developing robust defense mechanisms.

---

### Review · Reviewer_h95B · 2025-11-08

**Summary Of Contributions:**

The paper investigates the robustness of mechanistic interpretability techniques under adversarial model perturbations. The authors propose ProxPulse, a novel attack that simultaneously manipulates two types of feature visualizations. Furthermore, the authors introduce a new attack (CircuitBreaker) based on ProxPulse that reveals the manipulability of visual circuits, highlighting their lack of robustness. The effectiveness of these attacks is validated across a range of pre-trained models and evaluation metrics.

**Audience:**

Yes

**Audience Explanation:**

Yes. The discovery of visual circuits is interesting and potentially valuable for future research on robust interpretability and adversarial defense mechanisms.

**Broader Impact Concerns:**

The authors have reported the Broader Impact in the manuscript.

**Claims And Evidence:**

No

**Claims Explanation:**

Not entirely. The authors provide a large number of visualization results to support their claims. However, the overall readability of the paper is poor, which makes many of the arguments vague and difficult to follow. More details, please see "Requested Changes".

**Requested Changes:**

1. The readability of the abstract needs improvement.
a) Before the sentence “This paper addresses limitations in existing works by proposing a novel attack called ProxPulse”, the manuscript does not clearly specify what the limitations actually are. Moreover, the concept of an attack has not been introduced beforehand, so its sudden appearance makes the paragraph abrupt and confusing.
b) The goal of the manuscript is not stated clearly; the phrase “highlighting their lack of robustness” further blurs the intended goal of the study.
c) While the experimental setup is important, the abstract should include a concise summary of the main experimental findings, rather than merely describing the setup.

2. Similar issues exist in the Introduction. In the first two paragraphs, the central idea is that interpretability is important but may become unreliable under attacks. This easily gives the impression that the goal of this work is to defend against attacks to ensure reliability. However, the next paragraph abruptly shifts to proposing a stronger attack. The transition is therefore confusing. It is recommended to reorganize the introduction to focus more clearly on the limitations of existing attacks and to use those as the motivation for the proposed method.

3. Figure 1 is not intuitive. The caption — “The first row (resp. second row) shows the natural initial (resp. final) feature visualization and initial (resp. final) synthetic feature visualizations” — is unclear and needs to be rewritten for better readability.

4. Parts of words in Figures 14 and 32 are truncated.

5. According to Table 2, the proposed method does not seem particularly strong in terms of classification performance — the robust accuracy is almost the same as the baseline.

6. The main text lacks comparisons with other existing methods, especially those mentioned in the related work section.

7. There are minor spelling and consistency issues, such as “vision circuits” should be “visual circuits”; and inconsistent forms like “after fine-tuning” and “after finetuning”.

8. All ablation studies are placed in the appendix, which makes it difficult for the main text to independently support the core claims. The most important ablation results should be included in the main body of the manuscript.

9. Similar to Comment 2, both the introduction and the conclusion emphasize the importance of defense, yet the manuscript provides no discussion or preliminary analysis on this aspect.

---

> ### Author Response · Authors · 2025-12-28
>
> We thank Reviewer h95B for their constructive feedback on improving the paper's readability and presentation. We have carefully addressed each of your concerns and believe the revisions have significantly strengthened the manuscript. We detail our responses below.
>
>
>
> **1. Abstract and 2. Introduction Clarity**
>
> We appreciate your feedback on the abstract's structure and clarity. **We have completely rewritten the abstract to address all three concerns you raised**.
>
> First, we now explicitly introduce the concept of adversarial model manipulation early in the abstract, explaining that it involves "models being subtly perturbed to alter their interpretations while maintaining performance." This provides necessary context before discussing attacks.
>
> Second, we clearly enumerate the two key limitations of existing work before introducing ProxPulse: (1) existing methods manipulate either synthetic or natural feature visualizations individually, but not both simultaneously, and (2) no prior work has studied whether circuit-based interpretations are vulnerable to such manipulations. This makes the motivation for our work immediately clear.
>
> Third, we have clarified our goal by changing "addresses limitations" to "exposes these vulnerabilities," making it explicit that we aim to reveal weaknesses in current interpretability methods. We also now include concrete experimental findings in the abstract: "ProxPulse changes both visualization types with <1% accuracy drop while our CircuitBreaker attack manipulates visual circuits with attribution correlation scores dropping from near-perfect to ~0.6 while preserving circuit head functionality." This gives readers specific quantitative results rather than just describing the experimental setup.
>
> **For the introduction, we have added a new paragraph that explicitly states our framework and goal before discussing existing attacks**.  It aims to eliminate the confusion about whether we are defending or attacking, making clear that identifying vulnerabilities is a necessary step toward building robust interpretability methods.
>
> **3. Figure 1 Caption**
>
> The original caption was confusing. **We have completely rewritten it for clarity**:
>
> **4. Figure Quality Issues**
>
> We thank you for catching the truncation issues. **Figure 12 (now Figure 17 in the revision) has been updated, and the truncation issue has been resolved**. Similarly, **Figure 32 (now Figure 5 after reorganization) has also been corrected**.
>
> **5. Table 2 and Classification Performance**
>
> We understand the potential confusion here. The goal of our ProxPulse attack is not to improve or degrade classification accuracy, but rather to maintain performance while simultaneously fooling both types of feature visualization. This is by design, as specified in our threat model (Appendix A), where the attacker wants to obfuscate interpretations while keeping the model deployable with unchanged performance. The fact that accuracy remains nearly identical to the baseline demonstrates the success of our attack in this threat model—it shows we can completely change interpretations without any noticeable performance degradation that would alert an auditor. We have clarified this point in the revision.
>
> **6. Comparisons with Existing Methods**
>
> **We now present the comparisons with existing methods in the main text rather than in the Appendix.* Regarding our ProxPulse attack, we compare it against Nanfack et al. 2024's push-up and push-down attacks in Table 1. **We have also added a comparison against Geirhos et al. 2023's attack in the main text as Figure 2.b**. Since the Geirhos et al. attack only manipulates synthetic feature visualization, we compare against it specifically when evaluating changes in synthetic feature visualization. These comparisons demonstrate that our ProxPulse attack outperforms existing baselines.
>
> **7. Spelling and Consistency**
>
> Thank you for identifying these issues. We have performed a thorough search-and-replace to correct all instances of "vision circuits" to "visual circuits" and standardized all usage to "fine-tuning" throughout the manuscript.
>
> **8. Ablation Studies Placement**
>
> We agree that having all ablations only in the appendix weakens the main narrative. **We have now moved several important ablation studies to the main paper, including new Figure 5 showing ProxPulse results on DenseNet-201 and new Figure 6 showing ProxPulse results on ResNet-152**.
>
> **9. Defense Mechanisms Discussion**
>
> **We have now expanded the conclusion to include a substantive discussion of potential defense mechanisms grounded in our findings**
>
>
> We believe these revisions have substantially improved the paper's clarity.

---

### Review · Reviewer_rZ64 · 2025-11-22

**Summary Of Contributions:**

The paper studies the problem of feature visualization of the mechanistic interpretation of deep neural networks. Particularly, the paper aims to design attack algorithms that keep the underlying network's performance unchanged while fooling the interpretation methods. The paper first proposes ProxPulse, which is not able to fool the interpretation method. The paper then proposes CircuitBreaker on top of ProxPulse, which shows the manipulability of visual circuits.

**Audience:**

Yes

**Audience Explanation:**

Visual circuits and mechanistic interpretation are important directions to interpret and improve neural networks.

**Claims And Evidence:**

No

**Claims Explanation:**

The paper is overall clear, but I still have some concerns:

1. The objectives for ProxPulse and CirbuitBreaker are confusing. Firstly, C in eq 5 is described as a very large constant, and it is not clear how such a constant is chosen. Secondly, eq 5 uses log(1 + ) function, but it is not clear why such a design is adopted. Thirdly, ProxPulse is described to increase activations in the proximity ball, but I don't fully see why, by doing so, we can fool the interpretation method. Finally, in Eq. 6, the first term is described to maintain the feature visualization, which contradicts the goal.

2. The paper shows that ProxPulse fails to fool the interpretation methods while CircuitBreaker can. However, the paper does not provide much intuition for why this is the case or what it could indicate about the design's weaknesses or potential improvements of interpretation methods.

3. The evaluation of the robustness is mostly qualitative, i.e., through visual inspection of the identified synthetic images.

4. The paper has a limited focus on CNNs and does not seem to easily extend to ViT or other domains such as languages.

**Requested Changes:**

Please address the concerns above.

---

> ### Author Response · Authors · 2025-12-28
>
> We thank Reviewer rZ64 for their thoughtful feedback. We've made substantial revisions to address their concerns, which have significantly strengthened the paper.
>
> **1: Design choices for ProxPulse and CircuitBreaker**
>
> ** We've added detailed justifications in Section 4.1.**
> For the constant C, we set C=1e6 based on empirical observations—it's roughly 1000× larger than typical activation magnitudes we measured. This ensures the adversarial region we create dominates during gradient ascent. **We've also added sensitivity analysis in Appendix B showing our attack remains effective across C ∈ [1e5, 1e7], so the exact value isn't critical**.
> The log(1 + C/||·||²) formulation addresses two issues: numerical stability (preventing gradient explosion when activations approach zero) and providing smooth gradients for optimization. The 1/||·||² term ensures we target the smallest activations in the ball, which is key to creating a strong local maximum.
>
> **Regarding why this fools interpretation**: feature visualization finds inputs through gradient ascent, converging to the nearest activation maximum. By creating a high-activation region around our target images, we redirect this search—it now finds our adversarial region instead of the original learned patterns. This works for both synthetic (continuous optimization) and natural (discrete dataset) visualization.
>
> **For Equation 6, the first term maintains the circuit head's visualization, not the whole circuit**. We preserve the head so the circuit appears to detect the same high-level concept (e.g., "dog detector"), while the second term changes, which earlier-layer features contribute to this detection. The attack is stealthy, i.e., the output looks unchanged, but the internal mechanism is completely different. We've clarified this distinction in Section 4.2.
>
> **2: Why ProxPulse fails but CircuitBreaker succeeds**
>
> This is a great observation. We've added a new Section 5.4 analyzing this.
> **The key insight: ProxPulse changes individual neuron feature visualizations on a particular layer, but circuits use attribution scores averaged over the entire dataset**. These averaged gradients are robust to localized perturbations because changes near specific target images may get diluted when averaged over thousands of samples. Additionally, structured pruning selects entire kernel groups via L_0 regularization, making it less sensitive to neuron-level changes.
>
> CircuitBreaker works because it directly targets what circuit discovery uses, i.e., the attribution scores themselves through the pairwise ranking loss. Instead of **hoping activation changes indirectly affect rankings, we explicitly manipulate them**.
> This finding suggests an important design principle: interpretability methods based on multi-sample aggregation are inherently more robust than single-sample techniques. We discuss this in the conclusion as motivation for defense mechanisms.
>
> **3: Quantitative evaluation**
>
> We have extensive quantitative metrics that perhaps weren't sufficiently highlighted.:
>
> For ProxPulse: we used Kendall-τ, CLIP-δ, and pairwise CLIP similarities (Table 1, now with error bars from 5 seeds, and Figure 2).
> For CircuitBreaker: we used Pearson correlation, Kendall-τ on attribution scores, and similarity ratios (Figure 9, aggregated over 10 circuits).
>
> We've also moved key ablations (DenseNet, ResNet-152) to the main text. Visual inspection complements but doesn't replace these quantitative evaluations.
>
> **4: CNN focus**
>
> **We've added a Limitations paragraph acknowledging this**. "Our CNN focus is deliberate. Indeed, feature visualization and visual circuit discovery through structured pruning were specifically developed for CNNs. However, we acknowledge that extensions to other architectures require careful adaptation. For Vision Transformers, our framework could potentially use attention-based attribution scores instead of gradient-based ones, though circuit discovery methods for ViTs remain an open research question. For language models, recent work on activation patching and automated circuit discoveries suggests our framework could be applied, though the discrete nature of text inputs would require modifications to our proximity ball formulation. We view establishing these vulnerabilities in the well-studied CNN domain as an essential first step, with extensions to other architectures as important future work".

---

### Author Response · Authors · 2025-12-05
**Preparation of the response**

Dear Reviewers,
Thank you for your thoughtful and detailed reviews of our work. We appreciate the time and effort you've invested in evaluating our submission.
We are carefully reviewing all of your feedback and concerns. We will provide a comprehensive response addressing each point shortly.

Best regards,
The Authors

---

> ### Author Response · Authors · 2025-12-28
>
> **We sincerely thank all reviewers for their thoughtful and constructive feedback. The revisions have substantially strengthened the paper, and we believe it now addresses all concerns raised.**
>
> **Presentation and clarity**: Following Reviewer h95B's feedback, we completely rewrote the abstract to introduce the adversarial manipulation concept before discussing our attacks, explicitly enumerating the two key limitations of prior work, and including concrete quantitative findings rather than just describing the setup. **We also added a new introductory paragraph clarifying our goal upfront.**
>
> **Technical clarifications**: Reviewer rZ64 raised important questions about our design choices. We've added detailed explanations in Section 4.1 for why we chose C=1e6 (with sensitivity analysis showing robustness across values), why the log formulation prevents numerical instability, and how creating high-activation regions redirects gradient ascent in feature visualization.
>
> **Reproducibility**: Following Reviewer tX37's suggestions, **we've included our complete code as supplementary material** and will release a GitHub repository upon acceptance. We ran 5 independent seeds for all ProxPulse experiments—Table 1 now reports means and standard deviations. For CircuitBreaker, we aggregated results over 10 circuit heads with error bars. All hyperparameters are documented in Appendix B with sensitivity analyses.
>
> **Quantitative evaluation**: We've reorganized Section 5 to highlight our extensive quantitative metrics (Kendall-τ, CLIP-δ, Pearson correlations, similarity ratios) and moved key ablations from the appendix to the main text, including results on DenseNet-201, ResNet-152, and comparison with the Geirhos et al. baseline.
>
> **Defense and limitations**: We expanded the conclusion with a substantive discussion of potential defense mechanisms based on our preliminary experiments, and added an explicit limitations paragraph acknowledging our CNN focus while discussing how the framework could extend to ViTs and language models as future work.

---

### Decision · Action_Editor_383N · 2026-01-24

**Recommendation:** Accept as is

**Audience:**

Yes

**Audience Explanation:**

Research on mechanistic interpretability and its robustness is of interest to TMLR's audience.

**Claims And Evidence:**

Yes

**Claims Explanation:**

This paper proposes a new attack method on feature visualization called ProxPulse, and also proposes an attack method on visual circuits called CircuitBreaker. These attacks reveal vulnerabilities in feature visualization and visual circuit analysis, and experiments on ImageNet with several CNNs demonstrate their effectiveness.

The reviewers raised concerns about the paper’s writing clarity and its limited scope. However, based on the feedback, the authors revised the manuscript, and the clarity of both the writing and the scope has improved. The reviewers gave a positive assessment of the proposed attack methods and the experimental results.